# ROUND AND ROUND WE GO! ⟳ WHAT MAKES ROTARY POSITIONAL ENCODINGS USEFUL?

**Federico Barbero**[*]
University of Oxford

**Alex Vitvitskyi**
Google DeepMind

**Christos Perivolaropoulos**
Google DeepMind

**Razvan Pascanu**
Google DeepMind

**Petar Veličković**
Google DeepMind

## ABSTRACT

Positional Encodings (PEs) are a critical component of Transformer-based Large Language Models (LLMs), providing the attention mechanism with important sequence-position information. One of the most popular types of encoding used today in LLMs are Rotary Positional Encodings (RoPE), that rotate the queries and keys based on their relative distance. A common belief is that RoPE is useful because it helps to decay token dependency as relative distance increases. In this work, we argue that this is unlikely to be the core reason. We study the internals of a trained Gemma 7B model to understand how RoPE is being used at a mechanical level. We find that Gemma learns to use RoPE to construct robust 'positional' attention patterns by exploiting the highest frequencies. We also find that, in general, Gemma greatly prefers to use the lowest frequencies of RoPE, which we suspect are used to carry semantic information. We mathematically prove interesting behaviours of RoPE and conduct experiments to verify our findings, proposing a modification of RoPE that fixes some highlighted issues and improves performance. We believe that this work represents an interesting step in better understanding PEs in LLMs, which we believe holds crucial value for scaling LLMs to large sizes and context lengths.

## 1 INTRODUCTION

It is common to provide positional information to the attention mechanism in Transformers through the use of absolute positional encodings (Vaswani et al., 2017), relative positional encodings (Su et al., 2024), or by introducing a bias directly to the activations (Press et al., 2021). One of the currently most widely adopted encodings, especially in Large Language Models (LLMs), are Rotary Positional Encodings (RoPE) (Su et al., 2024), being used in popular models such as LLama 3 (Dubey et al., 2024) and Gemma (Gemma Team et al., 2024). RoPE acts on the queries and keys by splitting them in 2-dimensional chunks and rotating each chunk at a different frequency. The method can be implemented efficiently and provides an interesting geometric approach to positional encodings.

Despite the significant adoption of RoPE, the specific reasons why this method is useful to Transformer models remains poorly understood. One of the main arguments in favour of RoPE made by Su et al. (2024) is that the method helps to decay attention coefficients as the relative distance grows. Most such claims, however, rely on the queries and keys being *constant* vectors – which is uncommon in practice. In fact, in this work we find that there are many situations in which this decay does *not* occur and that this is exploited at times by attention heads in Gemma 7B (Gemma Team et al., 2024).

Further, there are open intriguing questions relating to how exactly the different frequencies in RoPE are useful. In the standard parameterisation of RoPE, the fastest frequencies rotate at 1 radian per token, whilst the slowest are several orders of magnitude slower at $\approx 1/10{,}000$ radians per token. As dot product attention directly depends on the angle between the queries and keys, the highest frequencies are extremely sensitive to small token rearrangements, making them poor carriers of

---

[*]Work performed while the author was at Google DeepMind.

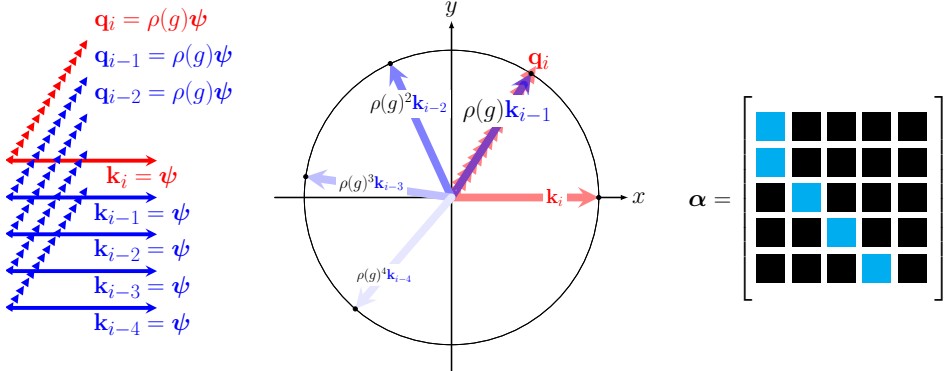

Figure 1: Depiction of our construction which allows Transformers to obtain positional attention heads using RoPE – zooming in on a single RoPE frequency for clarity. On the **left** we depict key and query vectors for each position $i$, where keys are all identical, and queries are just a rotated version of the key, in a way that matches one of RoPE's highest frequencies. The **center** depicts how keys get rotated by RoPE, making the key at $i-1$ perfectly align with the query. Due to the high frequency of the rotation, all other keys will lead to a smaller attention weight. On the **right** we show the resulting attention weights, resulting in this case in an off-diagonal positional attention. See Section 5 for more details.

information. Consequently, we find it interesting and important to understand how exactly the different rotation frequencies are used in LLMs.

**Contributions.** In this work, we study in-depth empirically and theoretically how Transformers and in particular auto-regressive LLMs benefit from RoPE. We rely on the pretrained and open-source Gemma 7B (Gemma Team et al., 2024) model for our empirical analysis and show consistent results with Llama3.1 8B (Dubey et al., 2024) in the Appendix (Section C). We summarise our findings:

- In Section 3, we argue against the common claim that RoPE is useful because it encourages the decay of attention coefficients with distance. We provide theoretical and empirical evidence to support our claim.

- In Section 4, we propose a new way to understand the usage of different frequencies in the queries and keys. We find that Gemma 7B largely prefers to use the low frequencies of RoPE. The first and last layers instead show the most use of the high frequencies.

- In Section 5, we show that the highest frequencies in RoPE are cleverly used by Gemma 7B to construct special 'positional' attention heads (see Figure 1). We mathematically prove the 'robustness' of the construction.

- In Section 6, we study how the low frequencies are used. We observe distinct 'bands' in the low frequencies of the queries and keys. We conjecture that Gemma 7B is using them as 'information channels'. We prove that these channels cannot be robust over long context.

- In Section 6.1, we propose a new technique called $p$-RoPE that removes the lowest frequencies of RoPE to create robust semantic channels. We show not only that removing a percentage of the frequencies maintains the performance, but also *improves it* on 2 billion parameter models. We believe that our work explains why increasing the maximum RoPE wavelength helps with long-context, e.g. as shown in Llama 3 (Dubey et al., 2024).

## 2 BACKGROUND

We denote by $\mathbf{x}_i \in \mathbb{R}^d$ the $d$-dimensional token embedding of the $i$-th token. Query and key vectors take the form $\mathbf{q}_i = \mathbf{W}_Q \mathbf{x}_i$ and $\mathbf{k}_i = \mathbf{W}_K \mathbf{x}_i$ respectively, given query and key matrices $\mathbf{W}_Q, \mathbf{W}_K \in \mathbb{R}^{d \times d}$. The attention mechanism[1] performs the following computation:

$$\alpha_{i,j} = \frac{\exp(\mathbf{a}_{i,j})}{\sum_{\ell \leqslant i} \exp(\mathbf{a}_{i,\ell})}, \text{with } \mathbf{a}_{i,j} = k(\mathbf{q}_i, \mathbf{k}_j) \tag{1}$$

---

[1]We ignore here the $1/\sqrt{d}$ scaling factor introduced by Vaswani et al. (2017) for ease of notation.

where $k$ is a kernel function which in our work takes the form of either a simple dot product, i.e. No Positional Encoding (NoPE) (Haviv et al., 2022; Kazemnejad et al., 2024), or RoPE. We focus on the case in which the attention mechanism is *causal*, the most common type of attention used in LLMs today. In a causal mechanism, we have that $\alpha_{i,j} = 0$ when $j > i$ and $\sum_{j:j \leqslant i} \alpha_{i,j} = 1$. We also note that we assume $\mathbf{a}_{i,j}$ to be finite, as it is always the case in practice, such that $0 < \alpha_{i,j} < 1$, when $j < i$. We call $\mathbf{a}_{i,j}$ the 'activation' or 'logit', while we call $\alpha_{i,j}$ the 'attention coefficient' between $i$ and $j$. It is sometimes useful to view the attention coefficients in matrix-form, which in our specific case results in a lower triangular row-stochastic matrix.

We highlight an important special case for $k$ which we call $k_{\text{NoPE}}$, i.e. no positional encoding: $k_{\text{NoPE}} (\mathbf{q}_i, \mathbf{k}_j) = \mathbf{q}_i^\top \mathbf{k}_j$, where $\mathbf{q}_i^\top$ denotes the tranpose of $\mathbf{q}_i$. In other words, in NoPE the kernel function computes simply the dot product, providing no positional information to the Transformer. It has been shown that Transformers can still perform well, especially out of distribution, with NoPE (Kazemnejad et al., 2024). In particular, Kazemnejad et al. (2024) prove that the Transformers could in principle recover absolute positional information through the causal mask; however, the proof relies on the universal approximation theorem, which we believe is a practical limitation.

## 2.1 ROTARY POSITIONAL ENCODINGS (RoPE)

For simplicity of notation in this work we assume that query are key vectors are $d$-dimensional, with $d \geqslant 2$ being an even number. We decompose queries and keys into 2-dimensional chunks $\mathbf{q}_i = \bigoplus_{k=1\ldots d/2} \mathbf{q}_i^{[k,k+1]} = \bigoplus_{k=1\ldots d/2} \mathbf{q}_i^{(k)}$, where $\bigoplus$ denotes direct sum (concatenation). In other words, we denote by $\mathbf{q}_i^{(k)} \in \mathbb{R}^2$ the $k$-th 2-dimensional chunk of the query vector of the $i$-th token, using analogous notation for the key vectors.

RoPE considers a sequence of angles $G = \left(g_k = \theta^{-2(k-1)/d} : k = 1, \ldots, d/2\right)$[2], where $g_1 = 1$ is the fastest rotating component at 1 radian per token and $g_{d/2} = \theta^{-(d-2)/d} \approx \theta^{-1}$ the slowest rotating component at approximately $1/\theta$ rotations per token. The parameter $\theta$ is called the base wavelength, which by default is 10,000 (Su et al., 2024), although works have explored increasing it to, for instance, 500,000 (Xiong et al., 2023; Roziere et al., 2023; Dubey et al., 2024). We denote by $\rho(g_k)$ the matrix form of $g_k$:

$$\rho(g_k) = \begin{bmatrix} \cos(g_k) & -\sin(g_k) \\ \sin(g_k) & \cos(g_k) \end{bmatrix}, \tag{2}$$

highlighting that $\rho(g_k)$ is a 2-dimensional orthogonal transformation (rotation). One can view $\rho(g_k)$ as a 'unit rotation' by $g_k$ radians. The RoPE technique amounts to the construction of a block-diagonal matrix $\mathbf{R}^i = \bigoplus_{k=1\ldots d/2} \rho(g_k)^i \in \mathbb{R}^{d \times d}$, where each $2 \times 2$ block on the diagonal is a rotation by a different frequency of RoPE. The $\mathbf{R}^i$ denotes in fact matrix exponentiation by an integer $i$ which is the position of $\mathbf{x}_i$[3]. We can exploit a nice property of rotation matrices, i.e. that $\rho(g_k)^i = \rho(ig_k)$ to avoid the computation of the matrix power. As this matrix is block diagonal, computing $\mathbf{R}_i \mathbf{q}_i$ means that the rotations act only on 2-dimensional chunks of the query (or key), i.e. $\mathbf{R}_i \mathbf{q}_i = \bigoplus_{k=1\ldots d/2} \rho(ig_k)\mathbf{q}_i^{(k)}$. This leads to the final formulation of $k_{\text{RoPE}}$:

$$k_{\text{RoPE}} (\mathbf{q}_i, \mathbf{k}_j) = \left(\mathbf{R}^i \mathbf{q}_i\right)^\top \left(\mathbf{R}^j \mathbf{k}_j\right) = \mathbf{q}_i^\top \mathbf{R}^{j-i} \mathbf{k}_j = \sum_{k=1\ldots d/2} \left(\mathbf{q}_i^{(k)}\right)^\top \rho(g_k)^{j-i} \mathbf{k}_j^{(k)}, \tag{3}$$

where we use the fact that $\left(\rho(g_k)^i\right)^\top \rho(g_k)^j = \rho(g_k)^{-i}\rho(g_k)^j = \rho(g_k)^{j-i}$. We highlight how the block diagonal structure of $\mathbf{R}$ allows one to decompose the dot product into the sum of dot products of 2-dimensional chunks, with each key vector chunk rotated at a frequency dictated by $g_k$.

## 2.2 RELATED WORKS

A number of works have investigated how different modifications of RoPE affect its generalisation. A well-known method involves increasing the parameter $\theta$ from the originally proposed 10,000 to a

---

[2]We denote the angles $g_k$ instead of $\theta_k$ as we opt for more of a group theoretic perspective of RoPE.

[3]For clarity, $i$ and $j$ have nothing to do with $\sqrt{-1}$, but instead denote the positions $i$ and $j$ of the tokens in the sequence.

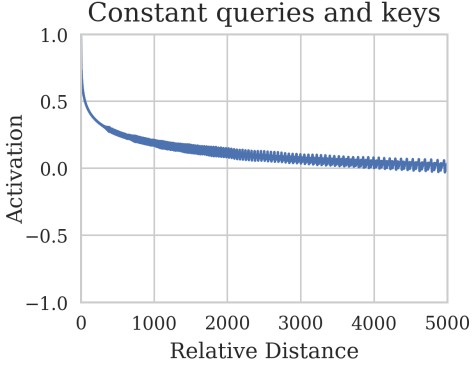
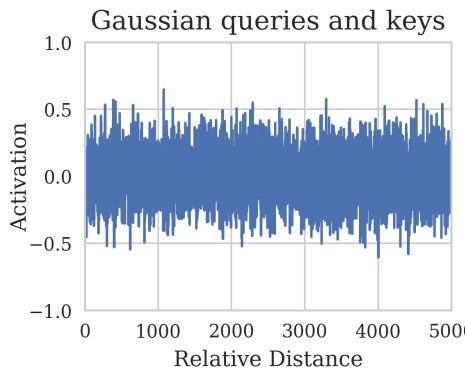

(a) RoPE applied to constant queries and keys.    (b) RoPE applied to Gaussian queries and keys.

Figure 2: RoPE applied to either (a) constant 'all-ones' queries and keys or (b) queries and keys with entries sampled IID from a Gaussian. The decay of the activations is present when the queries and keys are constant all-ones vectors, but not when they are Gaussian random vectors.

larger value, such as 500,000 (Xiong et al., 2023; Roziere et al., 2023). Notably, this is also used in the recently released LLama 3 class of models (Dubey et al., 2024). The justification provided for this modification is that a higher base wavelength means that the attention decay induced by RoPE will be slower, allowing for more robust learning over a larger context. In this work, we challenge this common assumption and investigate why RoPE and modifications such as these are helpful.

In a different direction, works have pointed out that NoPE shows strong performance in *out-of-distribution* (OOD) settings when compared to RoPE, arguing that the causal mechanism is sufficient to learn positional information (Haviv et al., 2022; Kazemnejad et al., 2024)[4]. In this work, we instead argue that NoPE and RoPE have *complementary* strengths and weaknesses and that for instance NoPE is unable to learn certain types of attention matrices present in Gemma 7B. Ruoss et al. (2023) propose to provide *randomised* positional information, showing that this helps boost OOD performance, claiming that this helps the model to learn over a longer range of relative distances. In our work (Appendix, Section B.3), we provide a different explanation as to why this kind of process might be helpful from the point of view of the type of invariance it encourages.

Overall, we believe this work provides a different and perhaps more nuanced perspective on different works that tackle positional encodings. In spirit, this work is similar to works that aim to understand LLMs from a mechanistic perspective (Elhage et al., 2021; Olsson et al., 2022; Wang et al., 2022; Hanna et al., 2024) and from the representations they produce (Barbero et al., 2024; Veličković et al., 2024). In the Appendix (Section B), we provide additional discussions on related works.

## 3    DOES ROPE DECAY ACTIVATIONS WITH DISTANCE?

In this section, we argue against the common claim that RoPE is helpful because it helps to decay activations as the relative distance between tokens increases. Such claims often work under the assumption that queries and keys are for instance a vector with all entries equal to each other, which we believe is an unrealistic oversimplification[5]. Importantly, this claim was originally provided by the authors of RoPE (Su et al., 2024) as a justification for the chosen structure of the encoding. Follow up works have used this claim to justify modifications, such as increasing the base wavelength $\theta$ to 500,000. We therefore find it important to point out cases in which this decay does not in fact occur.

We start by showing, in Proposition 3.1 that given any key, we can find a query such that RoPE is maximal for any chosen relative distance. This highlights the fact that RoPE provides Transformers

---

[4]Some works have argued that instead NoPE's extrapolation ability is still limited (Dong et al., 2024; Wang et al., 2024).

[5]Intuitively, Su et al. (2024) ask the question of what happens, as we vary the relative distance, to the dot product between already aligned queries and keys. This indeed will lead to a 'decay' with relative distance. However, this perspective ignores what happens to originally misaligned queries and keys, whose dot product can *increase* with relative distance.

with *robust* ways to attend to specific relative distances. In fact, we will show, in Section 4, that this mechanism is what Gemma 7B uses to construct heads that attend to specific positions. We provide the proof in the Appendix (Section A.1).

**Proposition 3.1** (RoPE can be maximal at arbitrary distance). *Given any query* $\mathbf{q}$ *and any relative distance* $r \in \mathbb{Z}$, *we can find a key* $\mathbf{k}$ *such that the softmax value is largest at distance* $r$ *with RoPE.*

Next, in Proposition 3.2, we show that given queries and keys sampled independently from a standard multivariate Gaussian, the expected value of the activations is 0. Moreover, this is independent of the relative distance of the queries and keys – implying that the expected value of the activations is independent of the relative distance when the queries and keys are sampled from a Gaussian. We provide the proof in the Appendix (Section A.1).

**Proposition 3.2** (Gaussian queries and keys do not decay.). *Let* $\mathbf{q}, \mathbf{k} \sim \mathcal{N}(\mathbf{0}, \mathbf{I})$. *Then, for any relative distance* $r \in \mathbb{Z}$, *we have that:*

$$\mathop{\mathbb{E}}_{\mathbf{q},\mathbf{k}\sim\mathcal{N}(\mathbf{0},\mathbf{I})} \left[ \mathbf{q}^\top \mathbf{R}^r \mathbf{k} \right] = 0.$$

We showcase this by constructing a synthetic experiment in which we either set queries and keys as all-ones vectors, e.g. as done by Su et al. (2024); Xiong et al. (2023), or sample entries independently from a Gaussian, accounting for appropriate normalisations. Figure 2 shows the results. While there seems to be some form of decay of the activations as relative distance increases when the queries and keys are all-ones vectors (a) – up to appropriate normalisation, this is clearly not the case when the queries and keys are instead random Gaussian vectors (b).

> **Summary of the Section:** *While RoPE helps to decay activations with relative distance in very specific conditions, this does not have to happen. In fact, we will see in the next section that this is something that Gemma 7B exploits to create specific attention patterns.*

## 4 HOW ARE DIFFERENT FREQUENCIES USED?

In this section, we explore how different frequencies of RoPE are used. RoPE relies on a set of frequencies $G$ that take values $1, \ldots, \theta^{-(d-2)/d}$, with the highest frequency varying by 1 radian per token, while the lowest being much more stable, varying at $\approx 1/\theta$ per token. As the angle between vectors affects the dot product, the contribution from the highest frequencies should behave similarly to random noise, i.e. a small perturbation in the token sequence will give a largely different activation contribution. A natural question is whether these frequencies are being used and, if so, how exactly are they helpful?

To measure the usage of frequencies, we start by noting that by Cauchy-Schwarz, the effect of the $k$-th frequency component on the activation $\mathbf{a}_{i,j}$ is upper bounded by the 2-norm of the query and key components, i.e. $|\langle \mathbf{q}_i^{(k)}, \mathbf{k}_j^{(k)} \rangle| \leqslant \|\mathbf{q}_i^{(k)}\|\|\mathbf{k}_j^{(k)}\|$. It is therefore natural to look at the mean 2-norm for each $\mathbf{k}^k$ in Gemma 7B over long sequences. For Gemma 7B, we note that $k = 1, \ldots, 128$, i.e. the hidden dimension is $128 \times 2 = 256$.

Figure 3 shows the results. We plot the average 2-norm at each layer of each rope 'chunk' over a number of sequences, ordering them by frequency, with the intuition that the norm will be an upper bound for how much that frequency will impact the activation dot product. We emphasize that the mean is taken over all of the 16 heads at each layer. It is clear that learning has assigned much higher norm on average to the lowest frequencies, meaning that they will likely influence the dot product the most. This seems to be true at each layer. Interestingly, there seems to be some high frequency usage present especially at the very first and last layers.

We believe this to be remarkable evidence showcasing how Gemma adapts to RoPE by preferring to use the lowest frequencies when computing attention activations.[6] In the Appendix (Section E.2, Figure 14), for completeness, we show that this type of distribution does not occur for the value vectors. This highlights that this behaviour is a consequence of RoPE, where learning discovers that these medium to high frequency 'chunks' are not useful and hence pushes their norm to 0 so that

---

[6]It remains under debate whether high attention scores imply a meaningful preference (Bibal et al., 2022).

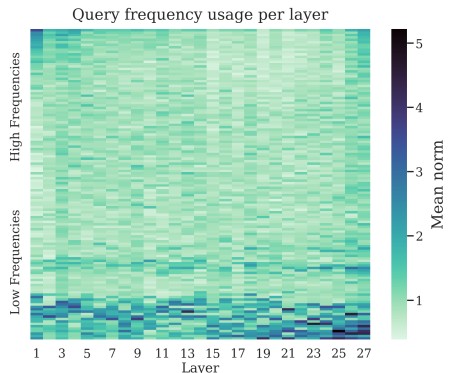 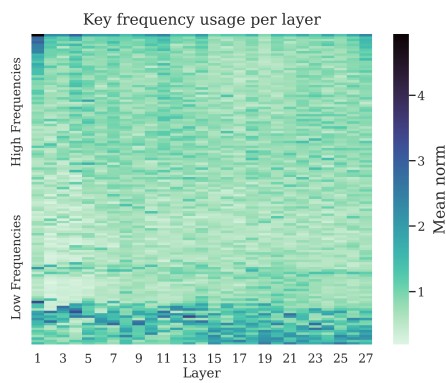

(a) Mean query norm distribution at each layer.     (b) Mean key norm distribution at each layer.

Figure 3: 2-norm plotted over 2-dimensional chunks of queries (a) and keys (b) for each layer in Gemma 7B, corresponding to different RoPE frequencies. A mean is taken over 10 different Shakespeare quotes and the 16 attention heads at each layer.

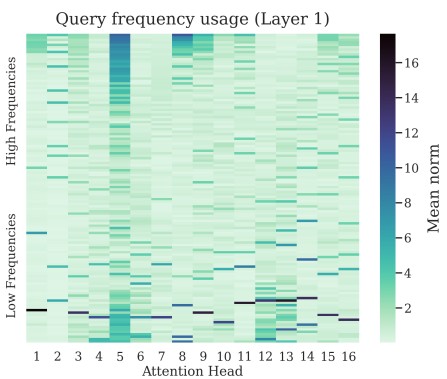 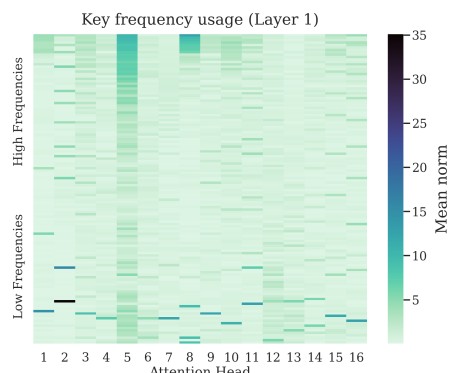

(a) Mean query norms at each attention head.     (b) Mean key norms at each attention head.

Figure 4: 2-norm plotted over 2-dimensional chunks of queries (a) and keys (b) for each attention head of the first layer in Gemma 7B, corresponding to different RoPE frequencies. A mean is taken over 10 different Shakespeare quotes. We explain in Section 5 the high frequency behaviour in Head 5 and Head 8.

their impact on the dot product is minimal. Corresponding entries in value vectors, not being rotated, do not suffer from the same pressure to have their norm 0.

In Figure 4, we show the frequency usage of the 16 attention heads in the first layer. The heads that stand out as using the high frequencies, especially for the keys, are Heads 5 and 8. We will show in the next section that these heads correspond to *positional attention heads*. We also highlight the sparse nature of the frequency usage, with the presence of 'high norm' bands, especially at the lower frequencies. We highlight that this kind of pattern seems consistent with the observation that feed-forward layers act as sparse dictionary lookup tables (Geva et al., 2021). In this context, we believe that these bands are used to perform some kind of sparse query and key semantic matching.

> **Summary of the Section:** *We empirically showed that most of the RoPE usage in Gemma 7B occurs at the low frequencies. We also identified 'high frequency' heads and high norm bands.*

## 5 HIGH FREQUENCIES: POSITIONAL ATTENTION

The highest frequencies in RoPE are interesting to study as their usefulness is not immediately obvious. In particular, in the previous section, we showed that Gemma 7B seems to largely avoid

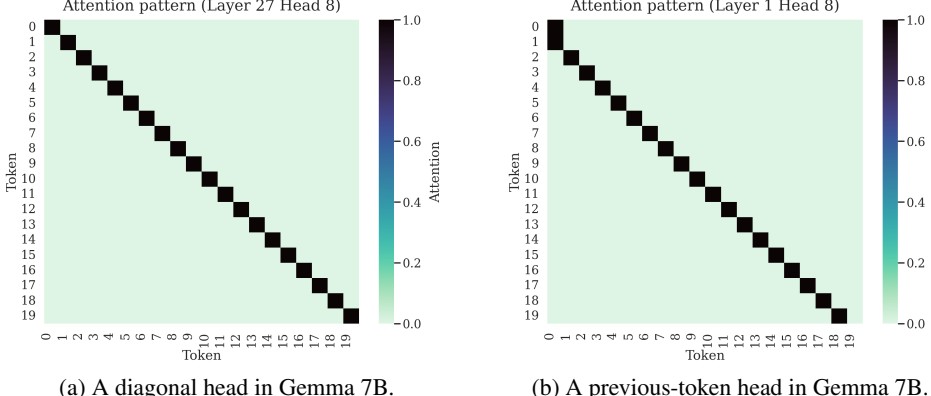

(a) A diagonal head in Gemma 7B.

(b) A previous-token head in Gemma 7B.

Figure 5: Examples of purely positional heads occurring in Gemma 7B, showcasing a diagonal head at the last layer (a) and a previous-token head at the first layer (b).

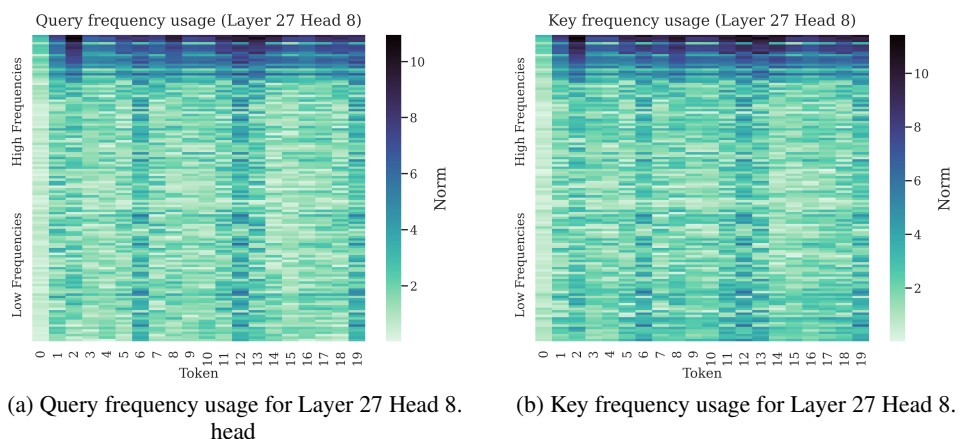

(a) Query frequency usage for Layer 27 Head 8. head

(b) Key frequency usage for Layer 27 Head 8.

Figure 6: Query and key frequency usage for a diagonal head present at the final layer, shown in Figure 5 (a). The head mostly relies on the high frequencies.

them. In this section, we will study the cases in which certain attention heads mostly use the very highest frequencies, showing that these heads tend to display purely positional attention patterns. We will prove by construction that RoPE allows for *arbitrarily sharp* attention patterns of this type – a construction that Gemma 7B seems to be closely learning in practice. We will also prove that such heads cannot be constructed with NoPE. *We believe the ability to construct such heads is important when trying to optimise for auto-regressive generation.* These heads have also shown to be useful for structure generalisation (see, e.g., Figure 6 in Ruoss et al. (2023)).

In Figure 5, we show certain heads in Gemma 7B that specialise into purely positional attention patterns – with Figure 5 (a) showing a 'diagonal' attention pattern and 5 (b) a 'previous-token' attention pattern. These heads seem to attend based purely on relative position, with no regards for semantics. It is not immediately clear how purely diagonal attention patterns are helpful, as these in principle behave like a residual connection. The reason *why* Gemma learns to construct such a pattern is outside the scope of this work, but constitutes interesting learnt behaviour which requires further analysis – perhaps a training 'bug'.

Figure 6, shows that for the diagonal head, Gemma seems to rely mostly on the very highest frequencies. We provide in the Appendix (Section E.2) more examples showing that this seems to be a repeated pattern throughout these positional heads. In fact, we found it remarkably consistent to identify positional heads, by looking at the usage of the highest frequencies. The heads 5 and 8 previously pointed in Figure 4 in fact correspond to positional heads with attention head 5 being purely diagonal and attention head 8 a previous-token head.

**Constructing robust positional heads.** We now study theoretically the mechanism through which Gemma 7B learns to robustly construct positional attention patterns. We start by defining more formally diagonal and previous-token attention patterns. Importantly, our definition relies on being able to learn patterns to an arbitrary precision $\epsilon$, given a maximum size $N$.

**Definition 5.1.** *An attention head can learn a diagonal attention pattern if given an $\epsilon > 0$ and a fixed size $N$, for all $i \leqslant N$, we have $\alpha_{i,i} > 1 - \epsilon$, implying that $\sum_{j:j<i} = \epsilon$. Similarly, for the previous-token head, we must have $\alpha_{i,i-1} > 1 - \epsilon$ for all $\epsilon > 0$, implying that $\sum_{j:j\neq i-1} \alpha_{i,j} = \epsilon$.*

We prove in Proposition 5.2 that no construction exists for NoPE that is able to learn such attention patterns. The proof relies on a counterexample involving token repetition and may be found in the Appendix (Section A.2).

**Proposition 5.2.** *An attention head with NoPE **cannot** learn a diagonal or off-diagonal pattern.*

Instead, in Theorem 5.3, we show that an attention head with RoPE is able to learn such patterns. The construction relies on setting queries and keys equal to each-other and for the off-diagonal case, inverse-rotating them by a unit rotation for each RoPE chunk. The proof is provided in the Appendix (Section A.2).

**Theorem 5.3.** *An attention head with RoPE **can** learn a diagonal or off-diagonal pattern.*

In fact, Gemma 7B seems to be learning a construction that is strikingly similar to our construction – we provide evidence in the Appendix (Section E.1). We also find why the highest frequency rotations are most similar, as they are the ones which will affect the dot-product most rapidly. For the diagonal case, setting queries and keys equal means that the dot product $\langle \mathbf{q}_i, \mathbf{R}^{j-i}\mathbf{k}_j \rangle$ will be maximal when $j = i$ and strictly smaller when $j \neq i$. The decrease in the dot product will depend on the angle between the rotated queries and keys. This angle will become dissimilar the fastest when using the highest frequency rotations. If $g_{d/2} \approx 1/10000$ for instance, then $g_{d/2}^{j-i}$ will misalign the queries and keys by a radian at a distance of $\approx 10,000$ tokens, while the highest frequency $g_1$ will only need a single token.

> **Summary of the Section:** *High frequencies in RoPE provide a mechanism to construct 'positional' attention patterns. NoPE is instead provably not able to construct such patterns.*

## 6 LOW FREQUENCIES: SEMANTIC ATTENTION

We now study how the low frequencies are being used. As observed in Section 4, the majority of the allocated norm (within the key and query embeddings) in Gemma 7B seems to be towards the lower frequencies. Furthermore, there seems to be a presence of distinct 'bands' in attention heads, especially in these low frequencies. In this section, we argue that the low frequencies are most useful to detect information related to token 'semantics' as they are the most invariant to token relative distance. The rotations are however not perfectly invariant over long relative distance, leading eventually to misalignment of the vectors also at the lowest frequencies.

In other words, the lowest frequencies are useful *as they are precisely the frequencies for which dot product is the least affected by the relative distance*. In particular, we believe this is why works such as LLama 3 have found it useful to increase the base wavelength to 500,000, meaning that the lowest frequencies rotate at roughly 1/500,000 radians per token. With a context length of 128k in LLama 3, the standard wavelength of 10,000 will in fact complete $\approx 2.04$ rotations, which might lead to poor generalisation due to tokens being erroneously misaligned simply due to the large context. We therefore deduce that with larger and larger contexts, the $\theta$ base wavelength of RoPE will have to also be increased accordingly – a relationship that has been studied in depth (Xu et al., 2024).

In Figure 7 (a), we identify an interesting semantic head that makes tokens appearing after an apostrophe token attend to the apostrophe[7]. We point out the qualitative difference when compared to the frequency usage in the positional heads from Section 5. This head is interesting being it is semantic (detecting apostrophes), but also positional, only detecting an apostrophe at the *previous*

---

[7]For instance, for the following 4 tokens [`BOS`, I, ', m], the attention head at token m will attend to the apostrophe token '. For all other tokens it will attend to the `BOS` token.

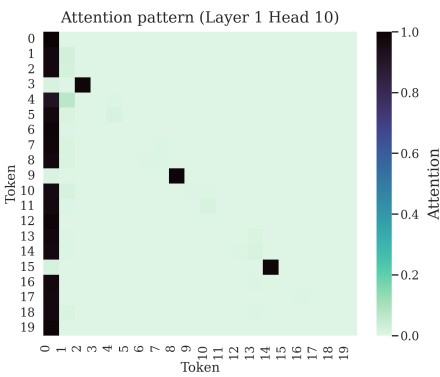
(a) Semantic 'apostrophe' attention head.

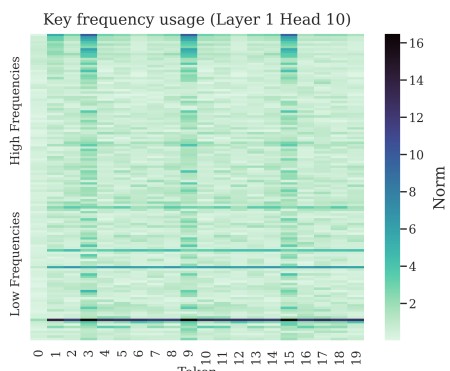
(b) Key frequency usage for the apostrophe head.

Figure 7: (a) Semantic attention head making tokens appearing after an apostrophe attend to it (b) Key frequency usage of same attention head, showing bands around low frequencies. The apostrophe tokens are at positions 3, 9, and 15.

*token*. We see how Gemma 7B is using the highest frequencies, visibly active in Figure 7 (b), to detect 'previous token' apostrophes. The low frequency dark band instead we believe is being used to attend robustly to the BOS token when there is no apostrophe present at the previous token – see the Appendix (Section E.1) for details.

We now however show that the 'semantic bands' cannot be robust over large context. In particular, irrational rotations are 'dense' meaning that given a long enough context, they can rotate to any arbitrary value. We use this to show that there exists a relative distance for which the attention activation will be erroneously large. We prove the 2-dimensional case, i.e. a single RoPE frequency, for simplicity. We believe that this is still an informative case as we observe that these bands are often very distinct and focused on a single frequency. We provide the proof in the Appendix (Section A.3) and point out that the observation that NoPE is instead able to build this type of attention pattern is immediate.

**Theorem 6.1.** *Let the hidden dimension be $d = 2$, i.e. there is only a single RoPE frequency $g_1$. Consider a sequence $\mathbf{x}_{bos}, \mathbf{x}_1, \ldots, \mathbf{x}_N$ with a target token at some position $\mathbf{x}_n$. Then for large enough $N$, the attention head with RoPE is unable to attend to such a token such that $\alpha_{i,n} > 1 - \epsilon$ for any $\epsilon > 0$ and any $1 \leqslant n \leqslant N$.*

## 6.1 TRUNCATING THE LOWEST FREQUENCIES OF ROPE

In the previous section and more formally in Theorem 6.1, we claimed that **RoPE lacks robust semantic channels**. To adapt to this, as shown in Figure 3, Gemma 7B seems to prefer to use the lowest frequencies to construct high norm semantic bands. The key insight is that unfortunately these high norm bands are not robust when the context becomes very large. We believe that this is why works such as Llama 3 found it helpful to increase the base wavelength of RoPE due to the larger context.

In this section, to investigate this further, we carry out an ablation in which we truncate the very lowest frequencies of RoPE. The intuition is that if our newly found intuition is correct, truncating the lowest frequencies *should not harm performance*. In fact, this allows RoPE to provide robust semantic channels that are distance agnostic. We call this modification $p$-RoPE, with $0 \leqslant p \leqslant 1$ being the fraction of RoPE 'kept'. We highlight two special edge cases: $p = 0$ coincides with NoPE, while the case $p = 1$ with RoPE. As such, the quantity $p$ behaves like an interpolant between the two, with higher values of $p$ being closer to RoPE and lower to NoPE. We provide additional details in the Appendix (Section D) and summarise the properties of the three types of encodings in Table 1.

We would like to note that $p$-RoPE is in spirit similar to the idea of increasing the wavelength of RoPE from 10,000 to 500,000, first proposed by Roziere et al. (2023) and later employed by LLama 3 (Dubey et al., 2024). The effect that increasing the wavelength has is that of increasing the amount of tokens required to destroy semantic information, leading to improved long-range performance. For a wavelength that is large enough compared to the context, the very lowest frequencies can act as a

Table 1: A summary of the discussed theoretically properties of RoPE, NoPE and $p$-RoPE.

| Encoding | NoPE | RoPE | $p$-**RoPE** |
|---|---|---|---|
| Positional | ✗ | ✓ | ✓ |
| Semantic | ✓ | ✗ | ✓ |

Table 2: Validation perplexity comparison between NoPE, RoPE and $p$-RoPE.

| Encoding | Wiki | FlanV2 |
|---|---|---|
| NoPE | 4.8594 | 6.6429 |
| $\text{RoPE}_{\theta = 10\text{k}}$ | 4.4627 | 6.4429 |
| $\text{RoPE}_{\theta = 500\text{k}}$ | 4.4485 | 6.4593 |
| $0.75\text{-RoPE}_{\text{reversed}}$ | 4.4592 | 6.4683 |
| $0.75\text{-RoPE}_{\text{partial}}$ | 4.4537 | 6.4562 |
| 0.25-RoPE | 4.5302 | 6.5111 |
| 0.75-RoPE | **4.4414** | **6.4422** |

semantic channel, as the position will have very little impact on the dot product. A summary of the properties between NoPE, RoPE, and $p$-RoPE is shown in Table 1.

We train Gemma 2B models from scratch on the `Wiki` and `FlanV2` training datasets (see Section D for details). We benchmark the use of RoPE, NoPE, and $p$-RoPE. We set the base wavelength $\theta$ to 10,000 and train and evaluate using the standard Gemma 8k token context. In Table 2, we show the perplexity on the validation set. We can see how truncating the lowest frequencies *not only maintains the same performance, but even seems to improve the validation perplexity*, supporting our claims regarding the low frequencies acting as non-robust semantic channels in standard RoPE. Truncating even more of the frequencies (0.25-RoPE) harms performance, but with results still significantly better than with NoPE. We find that $p$-RoPE also outperforms simply increasing the wavelength ($\theta = 500\text{k}$) and beats removing the highest frequencies (denoted $0.75\text{-RoPE}_{\text{reversed}}$) – validating our intuition that the lowest frequencies are the ones that should be removed. We further show that $p$-RoPE seems to outperform $\text{RoPE}_{\text{partial}}$ (see Appendix - Section D for a discussion).

Of course an 8k context is not very large and improvements due to wavelength increase typically were observed on a larger context of 32k (Xiong et al., 2023). We lack the resources to validate this on larger context lengths, but believe this to be a very interesting future direction. We in fact expect that truncating some small percentage of the lowest frequency tokens to help with long-context generalisation by providing robust semantic channels that are independent of relative distance and can therefore generalise automatically to any context length. We comment in the Appendix (Section B.5) further on this.

> **Summary of the Section:** *Low frequencies in RoPE tend to be used by semantic attention heads, as the rotation based on relative distance between tokens has minimal impact on the dot product. Removing the rotation for a certain percent of low frequencies improves performance on Gemma 2B models.*

## 7 CONCLUSION

In this work, we explored the behaviour of RoPE. We started by arguing that the use of RoPE does not ensure that attention weights will decay with relative distance. This is provably the case if we assume, for instance, that the keys and queries are Gaussian random vectors. Further, we provided a construction that ensure the attention weight is maximal at any provided relative distance, regardless of magnitude, with these observations going against the typical intuitions used to justify RoPE.

We provided evidence that high-frequencies bands are used to build circuitry that leads to positional attention. We provided a construction that ensures this behaviour and identified a strikingly similar construction within the attention heads of Gemma 7B. Furthermore we showed that NoPE is unable to do similar positional attention.

Finally we showed that low-frequencies are used for semantic attention, and argued that removing the RoPE rotation for these low-frequencies can potentially improve the ability to attend semantically and in particular to generalize such type of attention to longer contexts. We verified this through an ablation study with Gemma 2B. We overall see our work as providing a more nuanced understanding of RoPE and hope that it can lead to an improved understanding of how LLMs use positional encodings, ultimately leading to performance gains – especially over long contexts.

ACKNOWLEDGMENTS

The authors would like to thank Anian Ruoss (Google DeepMind), Simon Osindero (Google Deep-Mind), and Constantin Kogler (University of Oxford) for their valuable comments and suggestions on this work.

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

## A  PROOFS

In this section we provide the omitted proofs for the theoretical statements in the paper. We also provide more detailed discussions on the constructions and ideas behind the proofs. In Section A.1 we provide proofs for Section 3, in Section A.2 we provide proofs for Section A.2, and in Section 6 we provide proofs for Section A.3.

### A.1  DOES ROPE DECAY WITH DISTANCE?

In some of our proofs, we use known results on irrational rotations. We point to the literature for a detailed overview of these concepts (Masur & Tabachnikov, 2002), although the results we leverage in our work are elementary. The first statement is about the 'uniqueness' of irrational rotations.

**Lemma A.1.** *Consider $g \in \mathbb{Q}$ with $g \neq 0$ and $n \in \mathbb{Z}$. Then, $ng \equiv 0 \pmod{2\pi}$ only when $n = 0$. In particular, this also holds if $g$ is algebraic.*

We now prove that given a non-zero query $\mathbf{q}$ and a relative distance $r$, we can find a key $\mathbf{k}$ such that the activation is maximal at distance $r$. We note that the non-zero query assumption is minor as a zero query would imply with RoPE perfectly uniform attention as all activations would be equal to each other. The proof is by construction and involves rotating each $\mathbf{q}^{(k)}$ chunk by $\rho(g_k)^{-r}$. We then apply Lemma A.1 to show that the relative distance $r$ will give the maximum activation.

**Proposition 3.1.** *Given any (non-zero) query $\mathbf{q}$ and any relative distance $r \in \mathbb{Z}$, we can find a key $\mathbf{k}$ such that the softmax value is largest at distance $r$ with RoPE.*

*Proof.* Consider a distance $r$, a query $\mathbf{q} = \boldsymbol{\psi}$ that is non zero by assumption, and a key such that $\mathbf{k}^{(k)} = \rho(g_k)^r \boldsymbol{\psi}^{(k)}$ or equivalently $\mathbf{k} = \mathbf{R}^r \boldsymbol{\psi}$. Assume that the query is at position $i$ and the key at position $j \leqslant i$, we compute:

$$
\begin{aligned}
\mathbf{q}_i^\top \mathbf{R}^{j-i} \mathbf{k}_j &= \boldsymbol{\psi}^\top \mathbf{R}^{(j-i)+r} \boldsymbol{\psi} \\
&= \sum_{k=1\ldots d/2} \left(\boldsymbol{\psi}^{(k)}\right)^\top \rho(g_k)^{(j-i)+r} \boldsymbol{\psi}^{(k)} \\
&= \sum_{k=1\ldots d/2} \left\|\boldsymbol{\psi}^{(k)}\right\|^2 \cos\left((j-i+r)g_k\right).
\end{aligned}
$$

We now note that the maximum will only be achieved, by Lemma A.1, when $j - i = -r$, such that all $\cos\left((j-i+r)g_k\right) = \cos(0) = 1$, completing the proof. We note the minus sign coming from our convention that $j \leqslant i$, implying that $j - i \leqslant 0$. $\qquad\square$

We will later show that Gemma 7B seems to be learning a construction similar to the one above in practice in Section E.1. We next show that given queries and keys sampled from a standard

multivariate Gaussian, their expectation will be $0$. This follows from the fact that the standard multivariate Gaussian is invariant to rotations (isotropic).

**Proposition 3.2.** *Let* $\mathbf{q}, \mathbf{k} \sim \mathcal{N}(\mathbf{0}, \mathbf{I})$. *Then, for any relative distance* $r \in \mathbb{Z}$, *we have that:*

$$\underset{\mathbf{q},\mathbf{k}\sim\mathcal{N}(\mathbf{0},\mathbf{I})}{\mathbb{E}} \left[ \mathbf{q}^\top \mathbf{R}^r \mathbf{k} \right] = 0.$$

*Proof.* We will use the well-known fact that if $X \sim \mathcal{N}(\boldsymbol{\mu}, \boldsymbol{\Sigma})$ is a (multivariate) Gaussian random variable, then the linear transformation $Y = \mathbf{A}X + \mathbf{b}$ gives another Gaussian random variable such that $Y \sim \mathcal{N}(\mathbf{A}\boldsymbol{\mu} + \mathbf{b}, \mathbf{A}\boldsymbol{\Sigma}\mathbf{A}^\top)$. We start by computing:

$$\underset{\mathbf{q},\mathbf{k}\sim\mathcal{N}(\mathbf{0},\mathbf{I})}{\mathbb{E}} \left[ \mathbf{q}^\top \mathbf{R}^r \mathbf{k} \right] = \underset{\mathbf{q},\mathbf{k}\sim\mathcal{N}(\mathbf{0},\mathbf{I})}{\mathbb{E}} \left[ \sum_k \left( \mathbf{q}^{(k)} \right)^\top \rho(g_k) \mathbf{k}^{(k)} \right].$$

Let $\tilde{\mathbf{k}}^{(k)} = \rho(g_k)\mathbf{k}^{(k)}$. As $\mathbf{k}^{(k)} \sim \mathcal{N}(\mathbf{0}, \mathbf{I})$, we have that $\mathbb{E}[\tilde{\mathbf{k}}^{(k)}] = \rho(g_k)\mathbf{0} = \mathbf{0}$ and $\mathbb{V}[\tilde{\mathbf{k}}^{(k)}] = \rho(g_k)\rho(g_k)^\top = \mathbf{I}$ as $\rho(g_k)$ is orthogonal. This implies that also $\tilde{\mathbf{k}}^{(k)}$ is a standard Gaussian random variable, i.e. $\tilde{\mathbf{k}}^{(k)} \sim \mathcal{N}(\mathbf{0}, \mathbf{I})$. We proceed with the change of variables:

$$\begin{aligned}
\underset{\mathbf{q},\mathbf{k}\sim\mathcal{N}(\mathbf{0},\mathbf{I})}{\mathbb{E}} \left[ \sum_k \left( \mathbf{q}^{(k)} \right)^\top \rho(g_k) \mathbf{k}^{(k)} \right] &= \underset{\mathbf{q},\tilde{\mathbf{k}}\sim\mathcal{N}(\mathbf{0},\mathbf{I})}{\mathbb{E}} \left[ \sum_k \left( \mathbf{q}^{(k)} \right)^\top \tilde{\mathbf{k}}^{(k)} \right] \\
&= \underset{\mathbf{q},\tilde{\mathbf{k}}\sim\mathcal{N}(\mathbf{0},\mathbf{I})}{\mathbb{E}} \left[ \mathbf{q}^\top \tilde{\mathbf{k}} \right] \\
&= \sum_k \underset{\mathbf{q}_i\sim\mathcal{N}(0,1)}{\mathbb{E}} [\mathbf{q}_i] \underset{\mathbf{k}_i\sim\mathcal{N}(0,1)}{\mathbb{E}} [\mathbf{k}_i] = 0.
\end{aligned}$$

$\square$

### A.2 POSITIONAL ATTENTION PATTERNS.

We focus on the proofs regarding positional attention. We start by showing that NoPE cannot learn diagonal or off-diagonal attention. The proof is by counter-example, using a sequence that has repeated tokens.

**Proposition 5.2.** *An attention head with NoPE **cannot** learn a diagonal or off-diagonal pattern.*

*Proof.* We start by proving the diagonal head claim. Recall that a diagonal attention head satisfies $\alpha_{i,i} > 1 - \epsilon$ for any $\epsilon > 0$.

Consider the sequence $[\mathbf{x}_{bos}, \mathbf{x}_1, \mathbf{x}_1]$ that has a repeated token. We have that $\mathbf{a}_{3,3} = \langle \mathbf{q}_3, \mathbf{k}_3 \rangle = \langle \mathbf{q}_3, \mathbf{k}_2 \rangle = \mathbf{a}_{3,2}$. We compute the attention coefficient $\alpha_{3,3}$ :

$$\alpha_{3,3} = \frac{\exp(\mathbf{a}_{3,3})}{\exp(\mathbf{a}_{3,1}) + 2\exp(\mathbf{a}_{3,3})} = \frac{1}{\exp(\mathbf{a}_{3,1} - \mathbf{a}_{3,3}) + 2} < \frac{1}{2},$$

which contradicts the requirement that $\alpha_{3,3} > 1 - \epsilon$ for any $\epsilon > 0$.

We now focus on the second claim. Recall that an off diagonal head satisfies $\alpha_{i,i-1} > 1 - \epsilon$ for any $\epsilon > 0$. The same example contradicts this condition, as we have that $\alpha_{3,2} < \frac{1}{2}$. $\square$

We instead show that RoPE is able to learn such attention patterns robustly. The construction is related to the construction used in the proof of Proposition 3.1.

**Proposition 5.3.** *An attention head with RoPE **can** learn a diagonal or off-diagonal pattern.*

*Proof.* We start by proving the diagonal case. The construction involves setting all queries and keys equal, i.e. $\mathbf{q}_i = \mathbf{k}_j = \psi$ with $\psi$ not zero. To simplify notation and book-keeping, we focus on the case in which the hidden dimension is only $d = 2$, i.e. only acts through a single rotation. We will then explain how to generalise to $d > 2$. We call the rotation $g$ and its matrix representation $\rho(g)$.

We now note without RoPE that:

$$\langle \mathbf{q}_i, \mathbf{k}_j \rangle = \|\mathbf{q}_i\| \, \|\mathbf{k}_j\| \cos(\theta_{i,j}) = \|\psi\|^2 \,,$$

with $\theta_{i,j}$ the angle between $\mathbf{q}_i$, $\mathbf{k}_j$ which is 0 as we have by construction that $\mathbf{q}_i = \mathbf{k}_j = \psi$. When using RoPE, the dot product becomes:

$$
\begin{aligned}
\mathbf{a}_{i,j} &= \langle \rho(g)^i \mathbf{q}_i, \rho(g)^j \mathbf{k}_j \rangle \\
&= \left\| \rho(g)^i \mathbf{q}_i \right\| \left\| \rho(g)^j \mathbf{k}_j \right\| \cos\left( (j-i)\,g + \theta_{i,j} \right) \\
&= \|\psi\|^2 \cos\left( (j-i)\,g \right),
\end{aligned}
$$

where we use the fact that $\left\| \rho(g)^i \mathbf{x} \right\| = \|\mathbf{x}\|$ as $\rho(g)^i$ is an isometry. Due to Lemma A.1, we have that $(j-i)\,g \equiv 0 \pmod{2\pi}$ only when $j = i$. This implies that $\mathbf{a}_{ii} = \|\psi\|^2$ and $\mathbf{a}_{ij} < \mathbf{a}_{ii}$. We now compute $\alpha_{i,i}$:

$$
\begin{aligned}
\alpha_{i,i} &= \frac{\exp(\mathbf{a}_{i,i})}{\sum_{k<i} \exp(\mathbf{a}_{i,k}) + \exp(\mathbf{a}_{i,i})} \\
&= \frac{\exp\left( \|\psi\|^2 \right)}{\sum_{k<i} \exp\left( \|\psi\|^2 \cos\left( (k-i)\,g \right) \right) + \exp\left( \|\psi\|^2 \right)} \\
&= \frac{1}{1 + \sum_{k<i} \exp\left( \|\psi\|^2 \cos\left( (k-i)\,g \right) - \|\psi\|^2 \right)} \\
&= \frac{1}{1 + \sum_{k<i} \exp\left( \|\psi\|^2 \left( \cos\left( (k-i)\,g \right) - 1 \right) \right)}.
\end{aligned}
$$

We observe that $\cos\left( (k-i)\,g \right) < 1$ for $k \neq i$, implying that $\|\psi\|^2 \left( \cos\left( (k-i)\,g \right) - 1 \right) < 0$. In particular, we have that:

$$\sup_{\|\psi\|^2 \to \infty} \frac{1}{1 + \sum_{k<i} \exp\left( \|\psi\|^2 \left( \cos\left( (k-i)\,g \right) - 1 \right) \right)} = 1,$$

meaning that indeed our construction satisfies $\alpha_{i,i} > 1 - \epsilon$ for any $\epsilon > 0$, with $\epsilon$ a function of $\|\psi\|^2$.

We now focus on the off-diagonal case. Here we set all queries $\mathbf{q}_i = \psi$ and all keys $\mathbf{k}_i = \rho(g)\psi = \phi$, for $\psi$ non-zero. We now observe:

$$
\begin{aligned}
\mathbf{a}_{i,i-1} &= \langle \rho(g)^i \mathbf{q}_i, \rho(g)^{i-1} \mathbf{k}_i \rangle \\
&= \langle \rho(g)^i \psi, \rho(g)^{i-1} \rho(g) \psi \rangle \\
&= \langle \rho(g)^i \psi, \rho(g)^i \psi \rangle \\
&= \left\| \rho(g)^i \psi \right\|^2 \cos\left( (i-i)\,g \right) \\
&= \|\psi\|^2 .
\end{aligned}
$$

The same reasoning for the diagonal case then follows. For the high-dimensional case, it is sufficient to take $\mathbf{k}_i = \mathbf{R}\psi$ or equivalently $\mathbf{k}_i^{(k)} = \rho(g_k)\psi^{(k)}$. $\qquad \square$

In practice, Gemma 7B learns a construction very similar to that used in Theorem 5.3, as we will show in Section E. Further, as softmax tends to be rather 'sharp', the norm does not in practice have to become very large for the construction to be robust. This can be seen for example in Figure 6, where the chunk norms are not above 10, but the attention matrix allocates robustly attention on the diagonal.

**Can NoPE still learn positional patterns?** We clarify that our results prove that NoPE cannot learn diagonal or previous-token patterns given a *head operating in isolation*. This is interesting because we found for instance that Gemma 7B learnt such patterns already at the first layer. Our proof shows that this would be impossible to learn robustly with NoPE. It is interesting to study the expressivity of an attention head operating in isolation because it aims to capture the 'expressive efficiency' of a single attention head. Of course, here we make no claims that a sequence of attention heads cannot learn such patterns. In fact, Kazemnejad et al. (2024), prove that this should be possible, with an application of the Universal Approximation Theorem (Cybenko, 1989).

### A.3 SEMANTIC ATTENTION.

The next statement is a useful result on irrational rotations, namely that they are *dense* in $[0, 2\pi]$. Intuitively, this means that given an irrational rotation $g$, we can find an integer $n$ such that $ng$ (mod $2\pi$) is arbitrarily close to any $g' \in [0, 2\pi]$. The statement is commonly shown via a pigeon-hole argument.

**Lemma A.2.** *The subset $\{ng \pmod{2\pi} : n \in \mathbb{Z}\} \subset [0, 2\pi]$ is dense in $[0, 2\pi]$.*

We now show a simple, but useful Lemma regarding a required condition for 'sharpness', namely that the activation entering the softmax has to be the largest to have an attention coefficient larger than $1/2$.

**Lemma A.3.** *Let $\mathbf{a} \in \mathbb{R}^n$ be a sequence of $n$ activations entering a softmax such that $\alpha = \text{softmax}(\mathbf{a}) \in \mathbb{R}^n$. Denote by $\mathbf{a}_i, \alpha_i$ the $i$-th activation and attention coefficient, respectively. If $\mathbf{a}_i < \mathbf{a}_j$ for some $j$, then $\alpha_i \leqslant \frac{1}{2}$.*

*Proof.* We have that $\alpha_i = \frac{\exp(\mathbf{a}_i)}{Z}$ with $Z = \sum_k \exp(\mathbf{a}_k)$. We first note that if $\mathbf{a}_i < \mathbf{a}_j$, then $\alpha_i < \alpha_j$ as $Z$ is constant and $\exp$ is monotonically increasing. Now assume by contradiction that $\alpha_i > \frac{1}{2}$, we then have that $\alpha_j > \alpha_i > \frac{1}{2}$, implying that $\alpha_i + \alpha_j > 1$. This violates the sum-to-one constraint of the softmax function, implying that $\alpha_i \leqslant \frac{1}{2}$ if $\mathbf{a}_i < \mathbf{a}_j$ for some $j$. $\square$

We are now ready to show the result regarding the instability of RoPE. The intuition of the proof is that depending on the position in the sequence, the rotation can make the dot product positive or negative. This means that we can always find permutations, given a long enough sequence, that will make the activation of the desired element smaller than another element.

**Theorem 6.1.** *Let the hidden dimension be $d = 2$, i.e. there is only a single RoPE frequency $g_1$. Consider a sequence $\mathbf{x}_{bos}, \mathbf{x}_1, \ldots, \mathbf{x}_N$ with a target token at some position $\mathbf{x}_n$. Then for large enough $N$, the attention head with RoPE is unable to attend to such a token such that $\alpha_{i,n} > 1 - \epsilon$ for any $\epsilon > 0$ and any $1 \leqslant n \leqslant N$.*

*Proof.* Assume the desired element is at position $n$, that the head is correctly attending to it such that $\alpha_{i,n} > 1 - \epsilon$ for all $\epsilon > 0$ and $i \geqslant n$, and that $N$ is large enough. By Lemma A.3, the fact that $\alpha_{i,n} > 1 - \epsilon$ implies that $\mathbf{a}_{i,n} > \mathbf{a}_{i,j}$ for all $j$. As $\mathbf{a}_{i,n}$ has to be the largest element, we have two cases. Either $\mathbf{a}_{i,n} \leqslant 0$ is non-positive, implying that $\mathbf{a}_{i,j} < 0$ for all $j \neq k$, or $\mathbf{a}_{i,n} > 0$ is positive.

We start with the first case, where $\mathbf{a}_{i,n} \leqslant 0$. Consider an element at position $k \neq n$. We have that:

$$\mathbf{a}_{i,k} = \|\mathbf{q}_i\| \|\mathbf{k}_k\| \cos(\theta_{i,k} + g_1(k - i)),$$

where $\theta_{i,k}$ is the angle between $\mathbf{q}_i$ and $\mathbf{k}_k$. As $\mathbf{a}_{i,k} < 0$, we must have that $\theta_{i,k} + g_1(k - i)$ (mod $2\pi$) $\in \left(\frac{\pi}{2}, \frac{3\pi}{2}\right)$ (making the cosine negative). It is sufficient to *swap* the element at position $k$ with some element $k'$ such that $\theta_{i,k} + g_1(k' - i)$ (mod $2\pi$) $\in \left(0, \frac{\pi}{2}\right) \cup \left(\frac{3\pi}{2}, 2\pi\right)$, which is always possible for large enough $N$ by Lemma A.2. This swap would make $\mathbf{a}_{i,k'} > 0$, which by Lemma A.3, implies that now $\alpha_{i,k} \leqslant \frac{1}{2}$.

In other words, above we have shown that if an attention head can attend successfully at a certain position on a sequence that is long enough, certain swaps of the tokens in the sequence guarantee that this attention head would attend to the wrong element. In particular, if all activations are negative, it is sufficient to perform a swap that makes a non-target element positive.

We now focus on the case $\mathbf{a}_{i,n} > 0$. The principle is the same, but we need to apply two swaps. The first swaps $n$ with $k \neq n$, such that for the target token now at position $k$ we have that $\mathbf{a}_{i,k} < 0$ after the swap. If after the swap $\mathbf{a}_{i,k}$ is not the largest element anymore, we are done. If not, we need to perform a second swap with $k' \neq k'' \neq k$, which swaps an element at position $k'$ with $k''$ such that now $\mathbf{a}_{i,k'} > 0$. This forces the target element to not be the largest anymore. We can then apply again Lemma A.3 to complete the proof.

In other words, if some activations are positive, we need to perform two swaps. First we swap the target activation such that it becomes negative. Then, we are either done or we need to perform another swap making a non-target activation positive. These two cases imply the desired statement. □

## B ADDITIONAL DISCUSSIONS

In this section, we provide additional discussions omitted from the main part of the manuscript. We start by providing, in Section B.1 a more thorough discussion on related works. In Section B.2, we comment on how we believe our work and findings relate to positional encodings other than RoPE. In Section B.3, we comment on the work of Ruoss et al. (2023), which proposes a process that randomises the positions of tokens.

### B.1 ADDITIONAL RELATED WORKS

Our work is related to the field of mechanistic interpretability (Olah et al., 2020; Elhage et al., 2021) – whose focus is that of 'reverse engineering' models and more specifically in our case, LLMs. A number of works have studied and identified interesting behaviours of attention heads. Notable examples are induction heads (Olsson et al., 2022), chain-of-thought heads (Dutta et al., 2024), indirect object identification heads (Wang et al., 2022), multiple-choice heads (Lieberum et al., 2023), content-gatherer heads (Merullo et al., 2023), successor heads (Gould et al., 2023), attention sinks (Darcet et al., 2023), comparator heads (Hanna et al., 2024), and retrieval heads (Wu et al., 2024). Works have also exploited known patterns to speed-up inference time (Ge et al., 2023). In our work, we identify an 'apostrophe head' and 'positional heads' – as far as we are aware we are the first to identify the apostrophe head, while the diagonal and previous-token heads are instead likely to be already known.

While the identification of circuits and different types of attention heads constitutes an interesting direction, our work is conceptually different as it focuses on answering the question *how does RoPE help to construct such heads?* We believe this to be a rather new type of mechanistic interpretability approach as it is concerned with a 'lower level' of interpretability, in which one studies *how the Transformer implements* specific heads. In other words, we are more interested in the efficiency of the implementation rather than only limiting ourselves to understanding the high-level functionality. We believe that studies such as this one are important to better our understanding of the contributions of different components of LLMs, to ultimately improve them.

### B.2 OTHER POSITIONAL ENCODINGS

In our work, we focused on RoPE as we believe it is one of the most relevant positional encodings used today, being used for example by Gemma and Llama 3 models. We also find RoPE to have a particularly interesting geometric structure, that ties nicely with the dot product. These two reasons motivated our in-depth study of the method.

Regardless, we believe that our approach and findings can be transferred more broadly. First of all, we would like to point out the strong similarity between RoPE and the sinusoidal absolute encodings (APE) proposed by Vaswani et al. (2017), with both encoding position through a range of frequencies of sines and cosines. The fact that APE is added to the queries and keys and not applied as a linear map is a significant difference and would be interesting to explore in more detail. We suspect

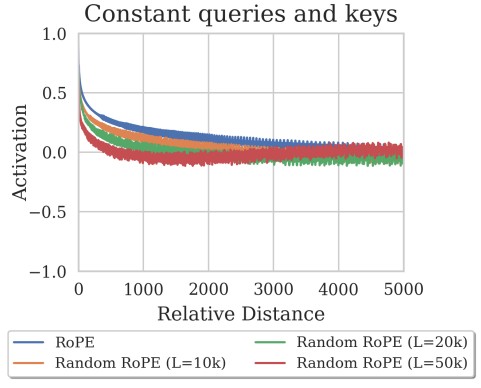 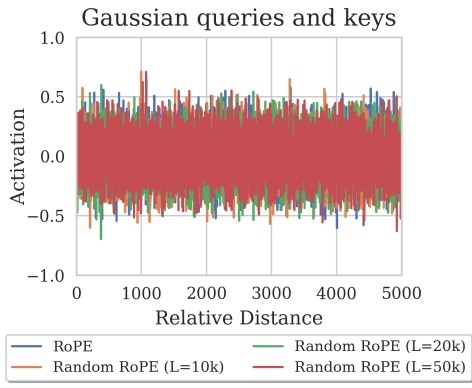

(a) RoPE and Random RoPE applied to constant queries and keys.

(b) RoPE and Random RoPE applied to Gaussian queries and keys.

Figure 8: RoPE and Random RoPE (Ruoss et al., 2023) applied to either (a) constant 'all-ones' queries and keys or (b) queries and keys with entries sampled IID from a Gaussian. We vary the $L$ parameter, showing that in (a), this tends to increase the decay rate.

however that our findings relating to the frequency usage to be somewhat transferable, with the lower frequencies being used to carry semantics and the high frequencies providing a method to encode positional attention patterns.

A popular method that is less similar to RoPE is Alibi (Press et al., 2021). We suspect that prior to this work, the common belief would be that Alibi and RoPE behaved in an analogous way, by decaying attention coefficients with distance. This is clearly true with Alibi. We however showed in this work that this is not the case with RoPE. Analysing Alibi in a way similar to this work would therefore be an interesting future direction of research to better understanding what kind of patterns these different positional encodings encourage. Overall, we hope that the understanding gained through our work will help the community to better classify different positional encodings.

### B.3 RANDOMISED POSITIONAL ENCODINGS

A recent work by Ruoss et al. (2023) proposes to 'randomise' the positions of RoPE, showing that this process helps to boost long-context generalisation performance. In particular, they propose given some maximum training context $N$, to choose some $L > N$, and to then sample without replacement from $1 \ldots L\ N$ elements, ordering them from smallest to largest, providing them as the positions during training and iteratively re-sampling. The motivation is that this process would allow the model to train using larger positions, helping with out-of-distribution generalisation to long sequences. We believe instead that this technique is instead helpful for a more subtle reason.

Providing during training random positions invites the model to learn some soft form of invariance to relative distance. The model has to in fact learn to produce the same output given different sampled positions when optimising the training loss. While pure invariance with RoPE cannot be achieved, the best the model can do when using RoPE is to rely on the very lowest frequencies to minimise the training loss. In other words, randomising the positions according to Ruoss et al. (2023) pushes the model to use the lowest frequencies. Testing this is laborious and outside the scope of our work, but we believe this to be an interesting future direction.

Furthermore, this randomised process has another side-effect, namely it increases the average relative distance. The effect of this increase in relative distance is that it will increase the decay of the activations. In Figure 8 (a), we show the effect of increasing the sampling upper bound $L$ on a constant vector of all-ones (up to normalisation). We see how increasing $L$ makes the decay occur faster, but that all curves seem to finally converge around 0. Instead in (b), we show that this does not affect the activation patterns when sampling from a Gaussian. This serves to counter the original claim that this process helps as the model will see larger relative distances. In fact, we see that as we increase the relative distance, the effect on the activations becomes more and more indistinguishable.

### B.4 ADDITIONAL RESULTS ON SYNTHETIC ACTIVATION DECAY

We provide in Figure 9 a further ablation where we show that two random Gaussian random vectors do not show decay properties. The difference between the plots from Figure 2 (b) is that here we do not sample a different Gaussian RV for each position in the sequence, but keep it constant. This provides supporting evidence for the fact that the decay visible in Figure 2 (a) is not only for constant queries and keys, but also that the queries and keys need to be 'perfectly aligned'.

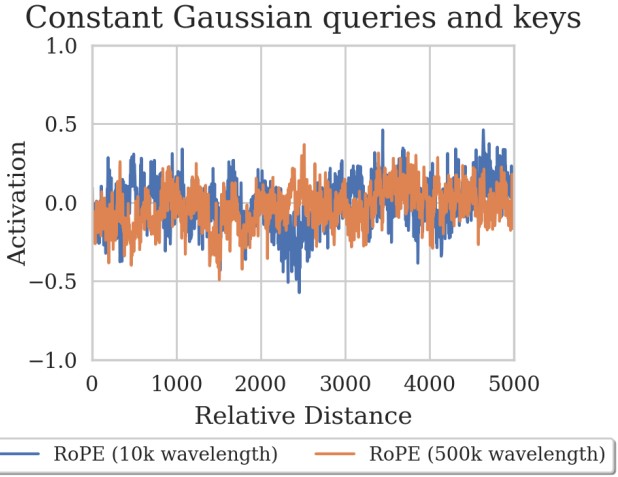

Figure 9: RoPE applied to 'constant' Gaussian random vectors. We sample a single Gaussian RV as a key and a single Gaussian RV as a query. We then repeat these vectors to form a sequence of length $n$. There is no clear evidence of decay.

### B.5 LIMITATIONS OF OUR WORK

As with other mechanistic interpretations, it is not clear that our description covers the entire spectrum of how LLMs use the RoPE encoding. While, the magnitude of the key entries corresponding to middle band of frequencies is considerably smaller on average, it is not $0$, and therefore it could still play a role. Our ablation of removing the low frequencies, while promising and helping to significantly validate our claims, could be improved with larger scale experimentation to asses the improvements specifically on large contexts – which we believe to be an important application of this work. Of course, such experimentation would be much more resource intensive.

Our current evaluation of $p$-RoPE involves the training of 2B parameter Gemma models from scratch and checking the perplexity on a validation set. Of course, lower perplexity does not necessarily mean better downstream performance (Kuribayashi et al., 2021). The goal of our work was to better understand RoPE. We found that our findings naturally lead to $p$-RoPE and we were happy to see that it showed strong initial signal. We leave to future work the more careful evaluation of $p$-RoPE as we believe this to be outside the scope of this paper.

## C LLAMA3.1 8B

In this section, we demonstrate that similar patterns also occur in the released LLama3.1 8B model (Dubey et al., 2024; Llama, 2024). The Gemma and Llama models we study are of course similar architecturally, but have significantly different design choices. For instance, Llama 8B uses Grouped Query Attention (GQA) (Ainslie et al., 2023) and uses a higher wavelength parameter of 500k. Further, the Llama3.1 models are trained with a much larger context of 128k tokens.

Figure 10 shows the results of the frequency analysis we report in Figure 3. We see that the queries (a) and keys (b) both seem to have higher activation towards the slower frequencies. Interestingly, we find that the start of the high norm bands occurs at approximately $500{,}000^{-40/64} \approx 0.0001$, which is roughly close to where they appear in the Gemma model with the much lower max wavelength

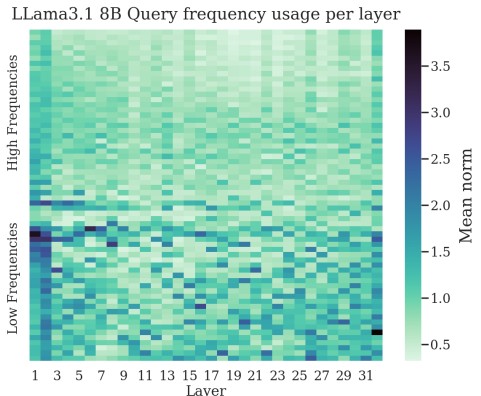

(a) Mean query norm distribution at each layer.

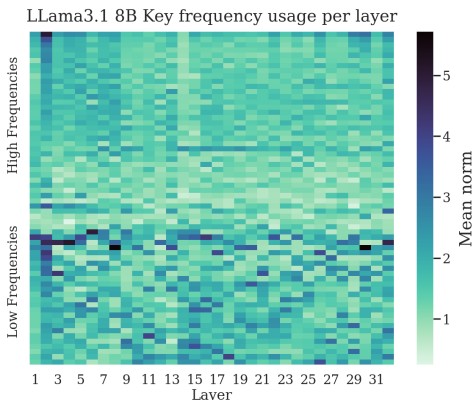

(b) Mean key norm distribution at each layer.



(c) Value key norm distribution at each layer.

Figure 10: Norm plotted over 2-dimensional chunks of Queries (a), keys (b), and (c) Values for each layer in Llama 3.1 8B, corresponding to different RoPE frequencies. The same process of frequency display does not show any clear pattern at each layer as RoPE is not applied to the values. This supports the claim that the patterns shown in the queries and keys are due to RoPE. The queries and keys seem to prefer to use the lower frequencies.

parameter at $10{,}000^{-0.8} \approx 0.0006$. This suggests that the effect of increasing this wavelength is to effectively provide to the model more 'slow enough' frequencies that seem to be useful for the model to use. As in the Gemma model, the value vectors do not have such a distribution of norms also in the Llama model.

We further provide evidence that the diagonal head pattern also occurs in Llama3.1 8B. In Figure 11, we show the frequency usage of 4 attention heads located at the 32nd (last) layer in the 4th head group. We highlight the 3rd head of this group that seems to use the high frequencies much more prominently. This head is in fact a diagonal attention head.

# D SUPPLEMENTARY $p$-ROPE EXPERIMENTAL DETAILS AND RESULTS

We start by highlighting that mechanisms similar to $p$-RoPE have been discovered – although we were unaware of this at the time of writing this work. An example is this GitHub Issue that suggests to use *partial rotary embeddings* (RoPE$_{\text{partial}}$): https://github.com/lucidrains/x-transformers/issues/40. Other works have also found RoPE$_{\text{partial}}$ to be useful (Black et al., 2022; Liu et al., 2024).

We believe RoPE$_{\text{partial}}$ and $p$-RoPE to be similar in nature as they provide parts of the queries and keys that are void of rotations. However, given our study, we believe $p$-RoPE to be better motivated.

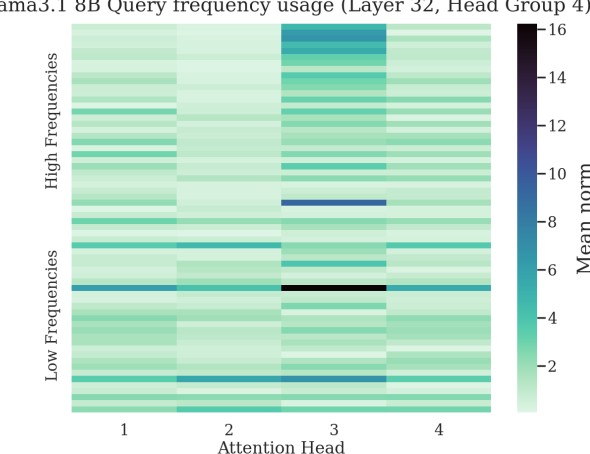

Figure 11: Frequency usage of the query vectors in LLama3.1 8B. We focus on the 32nd (last) layer and the 4th head group. Attention head 3, which has clearly more dominant high frequency usage, is a diagonal attention head.

Importantly, our study also helps to provide some insights into why a method such as partial-RoPE might be useful, which might be of independent interest regardless. We found in our experiments $p$-RoPE to outperform RoPE$_{partial}$.

In our additional experiments we train Gemma 2B from scratch separately on two datasets: English Wikipedia (`Wiki`) and `FlanV2`.

`Wiki` is a dataset based on English Wikipedia articles, built from the Wikipedia dump. Each sample contains the contents of a full Wikipedia article, with processing done to strip markdown and unwanted sections. The dataset is available at: [https://www.tensorflow.org/datasets/catalog/wikipedia](https://www.tensorflow.org/datasets/catalog/wikipedia). There are 6,672,479 documents and the total size is of 19.88 GiB. We held out 10% of the data randomly for the validation split.

`FlanV2` was introduced by Longpre et al. (2023). It is a dataset for instruction tuning which combines collections from FLAN, P3/T0, and Natural Instructions with dialog, program synthesis, and complex reasoning tasks. The dataset contains 15,000,000 samples.

In all experiments we train for 10,000 steps using a batch size of 512 and a sequence length of 8,192. We fix the wavelength to the standard value of 10,000. We use the standard Gemma 2B architecture (Gemma Team et al., 2024). For $p$-RoPE, we use the algorithm `apply_p_rope` described below.

```python
def apply_p_rope(
    inputs: jax.Array,      # [B, L]
    positions: jax.Array,   # [B, L]
    head_dim: int,
    max_wavelength: int = _MAX_WAVELENGTH,
    rope_percentage: float = 1.0,
) -> jax.Array:
  """Applies p-RoPE."""
  rope_angles = int(rope_percentage * head_dim // 2)
  nope_angles = head_dim // 2 - rope_angles

  fraction = 2. * jnp.arange(0, rope_angles) / head_dim
  timescale = max_wavelength**fraction
  timescale = jnp.pad(
      max_wavelength**fraction,
      (0, nope_angles),
      mode='constant',
      constant_values=(0, jnp.inf)
```

```
)

sinusoid_inp = (
    positions[..., jnp.newaxis] / timescale[jnp.newaxis, jnp.newaxis, :]
)
sinusoid_inp = sinusoid_inp[..., jnp.newaxis, :]
sin = jnp.sin(sinusoid_inp)
cos = jnp.cos(sinusoid_inp)

first_half, second_half = jnp.split(inputs, 2, axis=-1)
first_part = first_half * cos - second_half * sin
second_part = second_half * cos + first_half * sin
out = jnp.concatenate([first_part, second_part], axis=-1)
return out.astype(inputs.dtype)
```

## E  SUPPLEMENTARY GEMMA 7B RESULTS

We provide supplementary results for our analysis of Gemma 7B. We start by providing, in Section E.1, evidence supporting our claims regarding the types of constructions being learnt by Gemma 7B. We end by providing in Section E.2, the query, key, and attention pattern plots for all of the attention heads we study in the main part of the manuscript.

### E.1  CONSTRUCTIONS LEARNT BY GEMMA 7B

We explore the various constructions discussed in the main part of the manuscript. As a small aside for some geometric intuition, we note that the queries and keys in Gemma 7B are 256-dimensional. If they were randomly sampled from a Gaussian, their dot product should be very close to $0$. This follows from the fact that high-dimensional Gaussian random vectors are close to orthogonal.

**Diagonal heads.** We start by focusing on diagonal attention heads. In particular, we focus on the diagonal attention heads from Figure 14 and Figure 15. The hypothesis we want to verify is that the attention heads are learning to set queries and keys approximately equal to each-other, following the construction from Theorem 5.3.

To measure how close the queries and keys are, we leverage the Cauchy-Schwarz inequality. In particular, we know that:

$$\frac{1}{\sqrt{d}}\mathbf{q}_i^\top \mathbf{R}^{j-i}\mathbf{k}_j = \frac{1}{\sqrt{d}} \|\mathbf{q}_i\| \|\mathbf{k}_j\| \cos\left(\theta_{i,j} + g_{i,j}\right) \leqslant \frac{1}{\sqrt{d}} \|\mathbf{q}_i\| \|\mathbf{k}_j\|,$$

meaning that the quantity $\|\mathbf{q}_i\| \|\mathbf{k}_j\|$ serves as an upper-bound, achieved when the queries and keys are perfectly aligned. We highlight that it is important to introduce the normalisation factor $1/\sqrt{d}$ to get a meaningful bound. We then compare the activations (logits) to this upper bound. If the queries and keys are indeed aligned, the logits should be close to the upper bound. In Table 3 and Table 4, we showcase that this seems to be the case for the diagonal attention heads occurring at Layer 1 (Head 5) and Layer 27 (Head 8). The previous-token activations instead in comparison decay significantly in these heads.

**Previous-token head.** We perform a similar measurement for the previous-token head (Figure 16). An analogous situation occurs in which the previous-token activations seem to be much closer to the upper bound given by Cauchy-Schwarz, as seen in Table 5. We find that in comparison to the diagonal heads, the gap to the upper bound is larger. This is expected, as the head has to learn to model $\mathbf{R}$, which is a more complicated block-diagonal rotation matrix. Nevertheless, some tokens in particular are surprisingly aligned, supporting our claims.

**Apostrophe head.** We now analyse the construction of the 'apostrophe attention head' – see Section 6. In particular, we identified this head as one that attends to a previous-token apostrophe if it exists and to the BOS token, otherwise. We are particularly interested in understanding what the

| Token | BOS | 1 | 2 | 3 | 4 | 5 | 6 | 7 |
|---|---|---|---|---|---|---|---|---|
| Upper Bound | 3.74 | 182.34 | 203.65 | 483.97 | 141.48 | 130.71 | 168.52 | 122.31 |
| Diagonal | 3.25 | 119.60 | 156.93 | 383.00 | 98.47 | 92.15 | 137.10 | 90.71 |
| Previous-token | N/A | -8.45 | 76.67 | 82.13 | 29.01 | 45.33 | 73.04 | 55.25 |

Table 3: Activations for a diagonal head (Layer 1 Head 5). The upper bound is given by Cauchy-Schwarz. The diagonal activations are close to the upper-bound, meaning that the head is setting queries and keys to be very aligned to each other. The previous-token activations instead in comparison decay significantly.

| Token | BOS | 1 | 2 | 3 | 4 | 5 | 6 | 7 |
|---|---|---|---|---|---|---|---|---|
| Upper Bound | 7.91 | 64.42 | 112.43 | 51.97 | 52.69 | 67.10 | 116.41 | 67.37 |
| Diagonal | 6.05 | 55.84 | 96.52 | 45.75 | 44.00 | 53.83 | 105.47 | 57.78 |
| Previous-token | N/A | -9.63 | 32.51 | 29.49 | 25.96 | 20.08 | 19.95 | 6.07 |

Table 4: Activations for a diagonal head (Layer 27 Head 8). The upper bound is given by Cauchy-Schwarz. The diagonal activations are close to the upper-bound, meaning that the head is setting queries and keys to be very aligned to each other.

| Token | BOS | 1 | 2 | 3 | 4 | 5 | 6 | 7 |
|---|---|---|---|---|---|---|---|---|
| Upper Bound | 3.85 | 76.54 | 79.13 | 102.41 | 61.34 | 54.54 | 56.62 | 57.2 |
| Diagonal | -1.58 | 3.29 | 5.21 | -1.46 | 4.23 | 4.61 | 2.47 | 3.08 |
| Previous-token | N/A | 31.04 | 35.31 | 18.91 | 19.72 | 18.96 | 17.46 | 13.67 |

Table 5: Activations for a previous-token head (Layer 1 Head 8). The upper bound is given by Cauchy-Schwarz. While the diagonal activations seem to be close to 0, the previous-token activations are significantly larger and closer to the upper-bound.

low frequency band is being used for and how it acts as a semantic channel. We will show in this section that this semantic channel is being used to provide the BOS attention part of the pattern. In particular, this band is very positive when the token being attended to is the BOS token, and very negative otherwise. In Figure 12, we show the raw activations of the attention head. The column corresponding to the BOS token has activations of value $\approx 2$, while the apostrophe tokens on the off-diagonal token have attention values of $\approx 7$. The other values instead range from $\approx 0$ to the lowest of $\approx -12$, with all values being negative. These activations show how the head operates. It will default to attending to the BOS token, but will instead be dominated by an apostrophe token if it appears on the off-diagonal as it will have a more positive activation.

We now study the contribution from the low frequency, clearly visible in Figure 17 in the queries and keys. The semantic channel occurs at $g_{119} = 10{,}000^{-2(119-1)/256} \approx 0.0002$ – one of the lowest frequencies out of the 128 frequencies ($d = 256$) in Gemma 7B. As the context length is of $8k$, we get that the maximal rotation available is of $8000g_{119} \approx 1.64 \approx \frac{\pi}{2}$ radians, meaning that this channel is relatively stable amounting to a quarter of a full rotation, even at the largest context lengths. By Cauchy-Schwarz, as the norms corresponding to these frequencies are very large on average compared to the other frequencies, we expect this to 'dominate' the dot-product.

The queries corresponding to this frequency for the non BOS tokens are all set approximately to the same vector $\mathbf{q}_{not\text{BOS}}^{(119)} \approx [-4.1, 11.3]^{\top}$ and similarly the keys $\mathbf{k}_{not\text{BOS}}^{(119)} \approx [11.2, -3.5]^{\top}$. Instead, we have that $\mathbf{q}_{\text{BOS}}^{(119)} \approx [0.7, -1.9]^{\top}$, and $\mathbf{k}_{\text{BOS}}^{(119)} \approx [-2.5, 1.3]^{\top}$. We then note that $\left(\mathbf{q}_{not\text{BOS}}^{(119)}\right)^{\top}\mathbf{k}_{not\text{BOS}}^{(119)} \approx -85.5$, while $\left(\mathbf{q}_{not\text{BOS}}^{(119)}\right)^{\top}\mathbf{k}_{\text{BOS}}^{(119)} \approx +24.9$. In other words, this channel contributes a very negative activation for tokens attending to non-BOS tokens, and very positive activations to tokens attending to the BOS token. As the the underlying frequencies are very low, this mechanism will be stable given the limited context length of Gemma 7B. We however expect this mechanism to break over long enough sequences, in accordance with Theorem 6.1. This provides evidence supporting our claims that the lowest frequencies are used as semantic channels, as this kind of behaviour will be most robust exactly for those channels that provide the lowest frequencies.

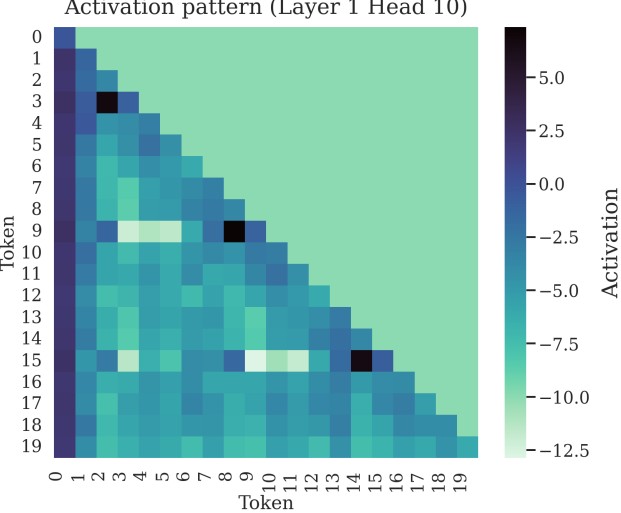

Figure 12: Activation pattern of the 'apostrophe head' in Gemma 7B. The upper triangular mask is artificially set to the value of $-10$ for clarity of the visualisation, but in reality takes a value of $\approx -10^{30}$. Token 0 is the BOS token. The 3 darkest activations on the off-diagonal correspond to tokens after an apostrophe token attending to the apostrophe token.

## E.2 ANALYSIS OF PATTERNS IN DIFFERENT ATTENTION HEADS

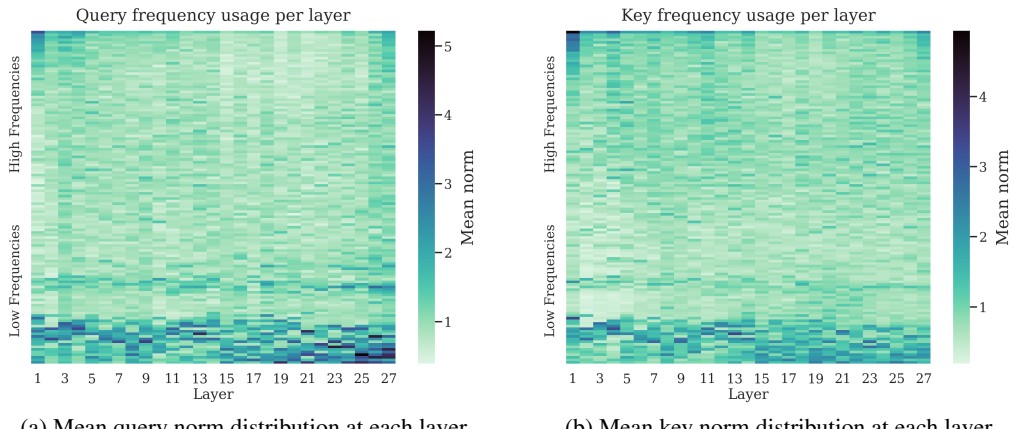

(a) Mean query norm distribution at each layer.     (b) Mean key norm distribution at each layer.

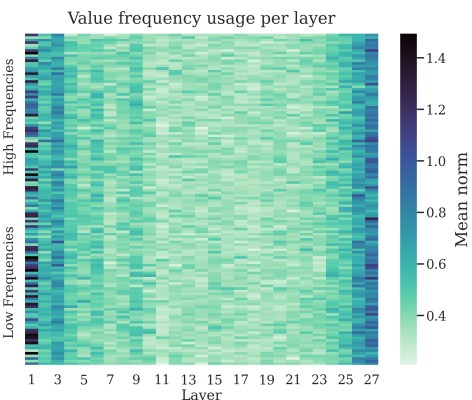

(c) Value key norm distribution at each layer.

Figure 13: Norm plotted over 2-dimensional chunks of Queries (a), keys (b), and (c) Values for each layer in Gemma 7B, corresponding to different RoPE frequencies. The same process of frequency display does not show any clear pattern at each layer as RoPE is not applied to the values. This supports the claim that the patterns shown in the queries and keys are due to RoPE.

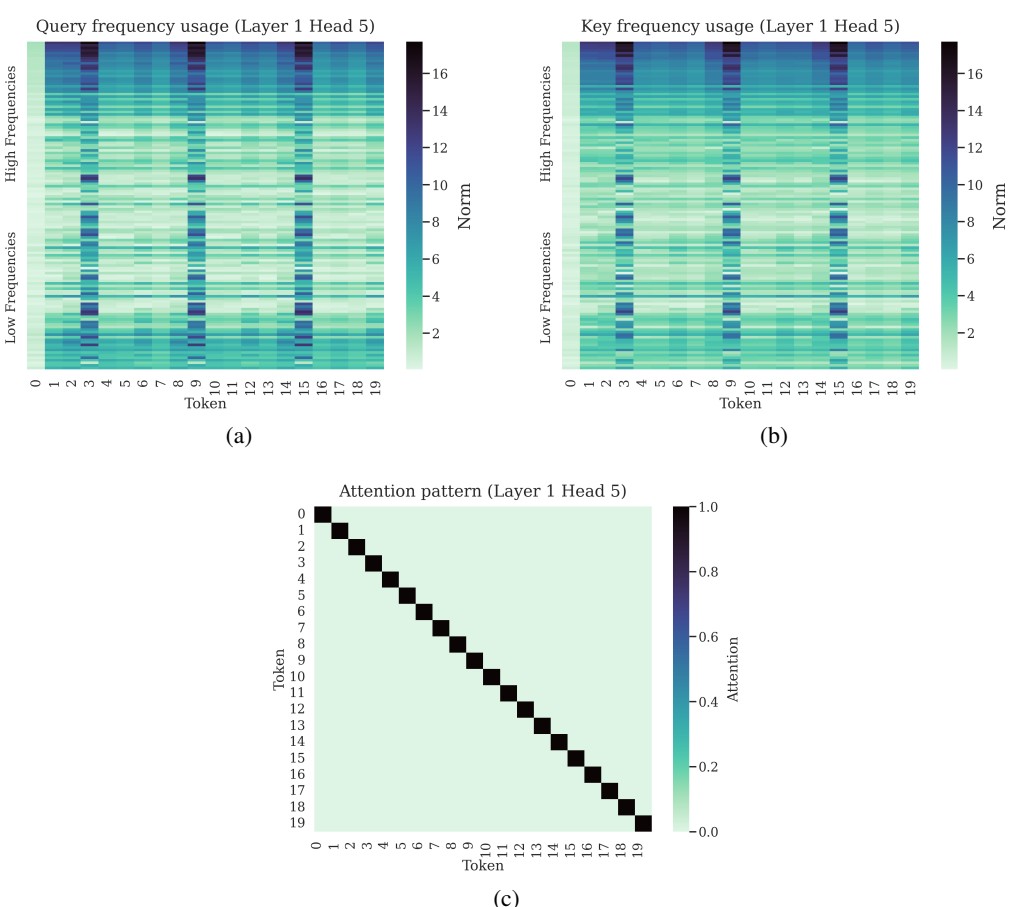

Figure 14: Queries (a), keys (b), and attention pattern (c) for a diagonal head present at the first layer in Gemma 7B. The head implements a 'residual connection'.

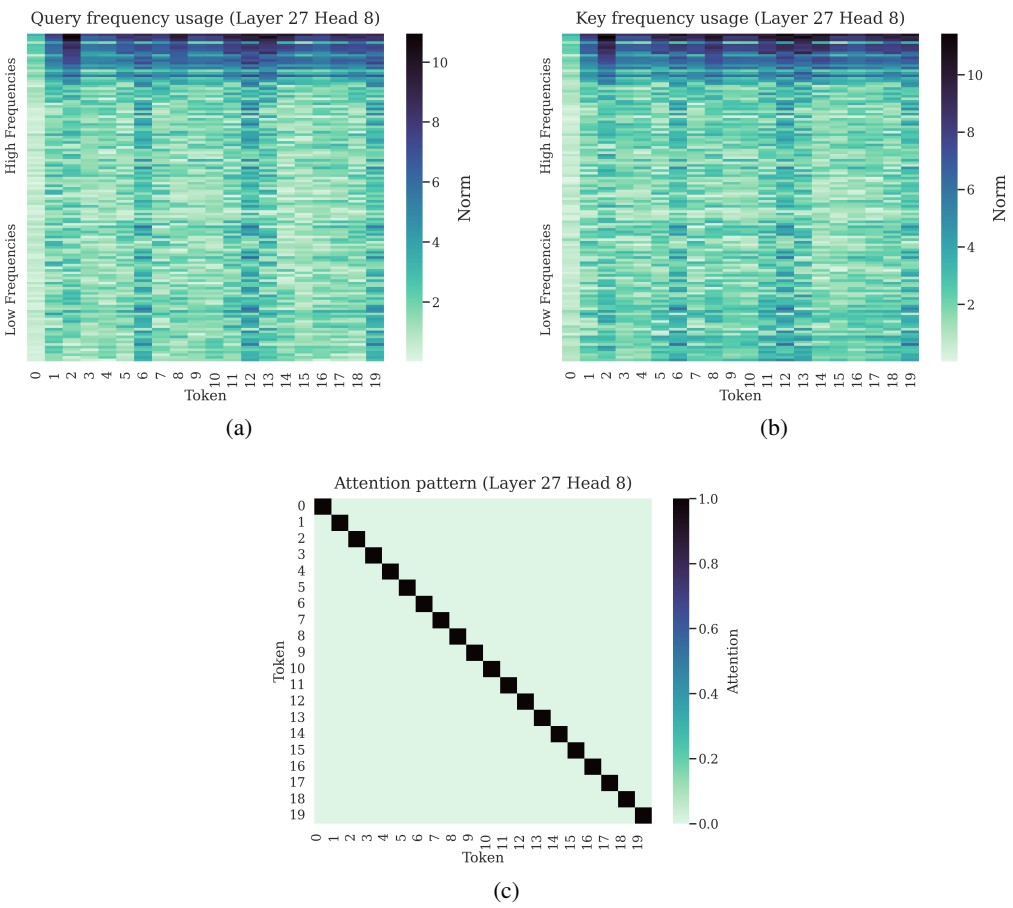

Figure 15: Queries (a), keys (b), and attention pattern (c) for a diagonal head present at the last layer in Gemma 7B. The head implements a 'residual connection'.

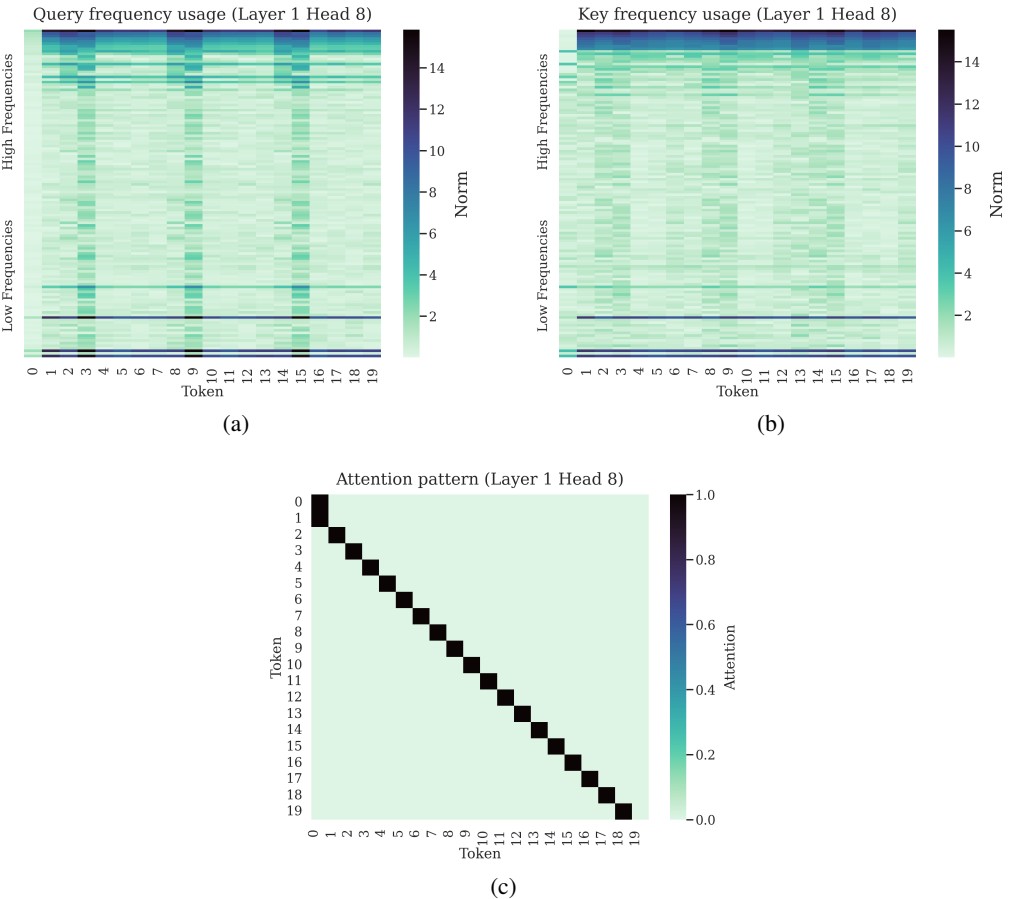

Figure 16: Queries (a), keys (b), and attention pattern (c) for the previous-token head in Gemma 7B. The head makes tokens attend to tokens appearing immediately before them.

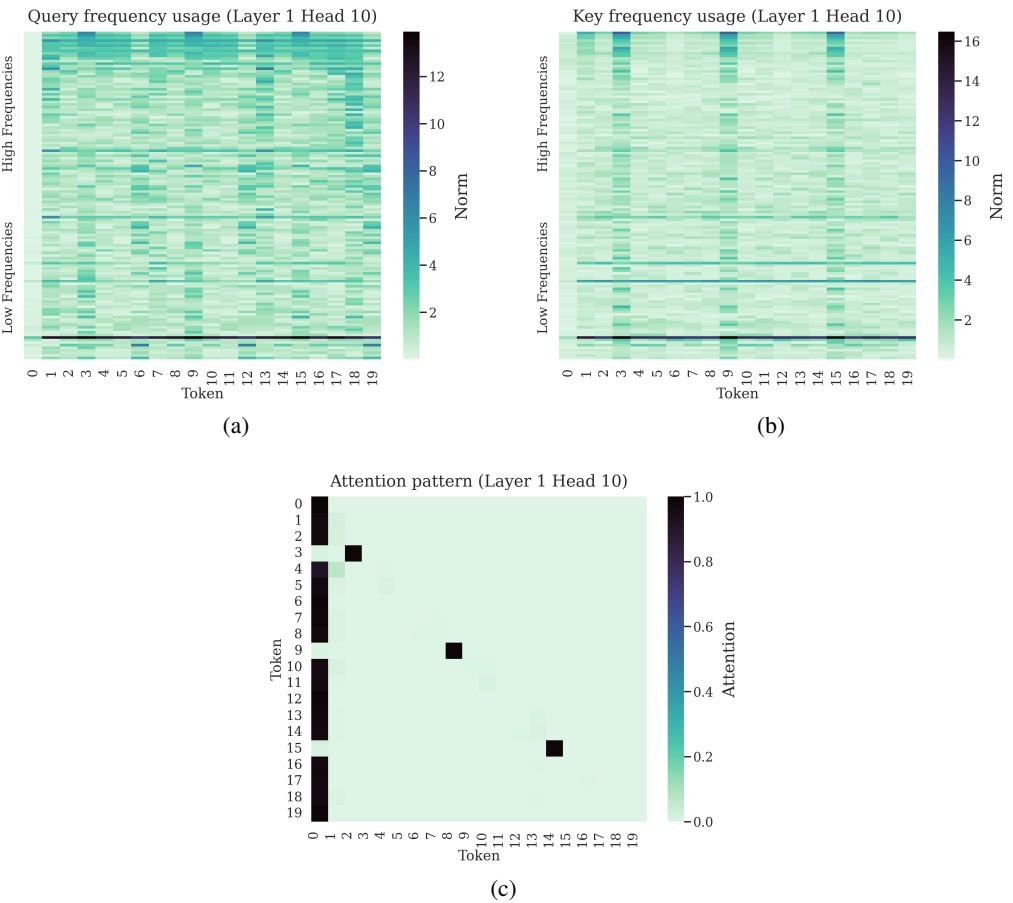

Figure 17: Queries (a), keys (b), and attention pattern (c) for the apostrophe head in Gemma 7B. The attention head learns to make tokens appearing after an apostrophe token attend to the apostrophe.

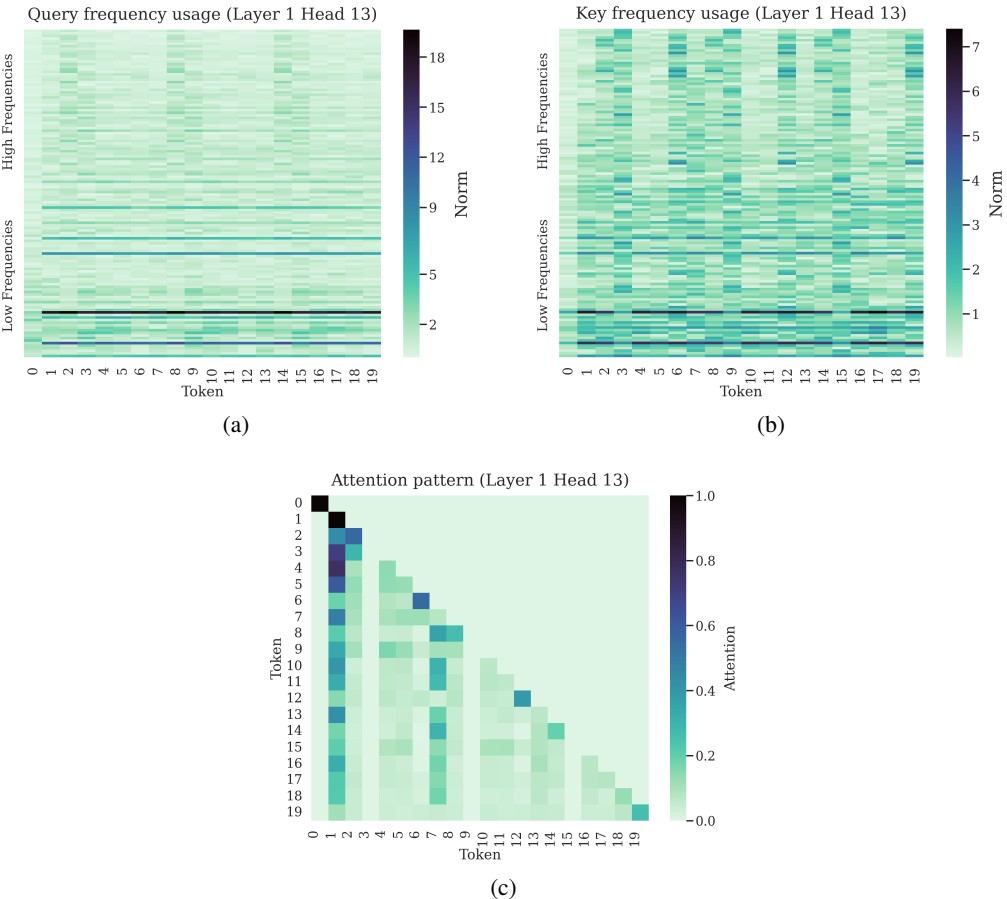

Figure 18: Queries (a), keys (b), and attention pattern (c) for a general 'semantic' attention head in Gemma 7B. The queries in particular display high norm bands on the low frequencies.

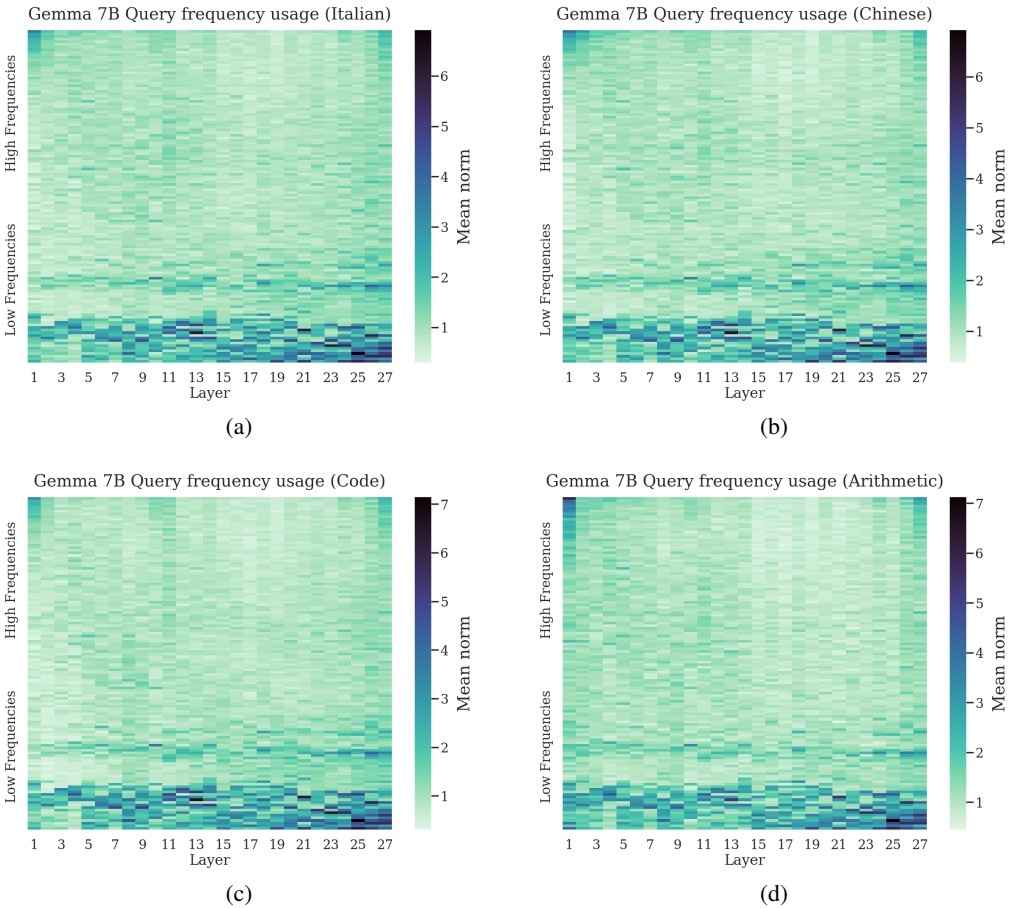

Figure 19: Query frequency usage for Gemma 7B over different input domains. (a) Shows the usage for Italian prompt, (b) for a Chinese prompt, (c) for Python Code, and (c) for Arithmetic additions between two large integers. The frequency usage patterns are very similar across all domains, with very slight differences.

