# OpenReview forum: "Round and Round We Go! What makes Rotary Positional Encodings useful?"
_ICLR.cc/2025/Conference — ICLR 2025 Poster_

### Official Review · Reviewer_DtVE · 2024-10-16

**Soundness:** 3
**Presentation:** 3
**Contribution:** 2
**Rating:** 5
**Confidence:** 3

**Summary:**

This paper studied the inner workings of rotary positional embedding. The authors started by challenging the common belief that RoPE decays with distance. Then, the authors showed in Gemma 7B that most RoPE usages appear in low frequencies. The authors explained that high frequencies are for positional information while low frequencies are for semantic information. Finally, the authors observed that the low-frequency components are not robust. Based on this observation, the authors proposed p-RoPE to remove (1-p)*100% of the low-frequency component and showed that this improved the performance of Gemma 2B models.

**Strengths:**

* The paper is well-written and the key points are delivered clearly.
* The authors made a good point that RoPE doesn't necessarily decay with distance.
* The figures clearly showed that most attentions happen in low frequencies, and some high-frequency heads and frequency bands exist.

**Weaknesses:**

* The claim that high-frequency components are for positional information is unclear.
  - The reasoning seems to be: (a) RoPE can learn a diagonal or off-diagonal pattern. (b) NoPE cannot learn a diagonal or off-diagonal pattern. So RoPE can learn positional information. However, it doesn't necessarily mean the high-frequency components contribute to the diagonal/off-diagonal pattern. So the function of high-frequency components remains unclear.
  - Some prior works provide evidence that NoPE can still encode positional information [1][2]. So the fact that NoPE cannot learn a diagonal or off-diagonal pattern may not imply it cannot encode positions.
* The authors experimentally show that truncating the lowest frequencies can help RoPE learn better. However, it is based on a claim that "RoPE lacks robust semantic channels".
  - This claim comes from an analysis on a 2-dimensional case. However, it is possible that the 2D case is too restrictive and a higher dimensional case could have a different result.

[1] Haviv et. al. "Transformer language models without positional encodings still learn positional information," EMNLP 2022 Findings.
[2] Chi et. al. "Latent positional information is in the self-attention variance of transformer language models without positional embeddings," ACL 2023.

**Questions:**

* Is your p-RoPE having a similar flavor as the partial rotary?
  - Partial rotary means the whole dimension is divided into rotary part and non-rotary part. The rotary part will be treated by RoPE while the non-rotary part is treated by NoPE.
  - Partial rotary embedding has been known by the community for a while (see https://github.com/lucidrains/x-transformers/issues/40). It is known that the partial rotary is slightly better than rotary positional embedding.
  - Because the proposed p-RoPE is interpolating between NoPE (p=0) and RoPE (p=1), it is possible that p-RoPE is similar to partial rotary.

---

> ### Author Response · Authors · 2024-11-18
>
> We are happy to hear that you found our work well-written and clear. We are also happy that you appreciated one of our important claims in the paper which provides evidence against the common claim present in important works that RoPE helps to decay with distance the signal. We would like to answer your questions and comments.
>
> **The claim that high-frequency components are for positional information is unclear...However, it doesn’t necessarily mean the high-frequency components contribute to the off-diagonal pattern. So the function of high-frequency components remains unclear.**
>
>
> We thank you for the great comment. In our paper we mathematically explain why the highest frequencies are the most effective in constructing the positional attention patterns, but indeed we do not make claims that they are *necessary*. What we instead claim is that given some fixed budget of vector “norm” the highest frequencies are precisely the most useful to have the sharpest patterns. For the details see the paragraph below Theorem 5.3 which explains exactly why the highest frequencies are the most useful when constructing these *sharp* attention patterns. Finally, we also left a discussion of this in the Appendix E.1.
>
> In practice, we believe that we provide ample evidence for this behaviour in our work – see e.g. Figure 14, Figure 15, and Figure 16, where we show how the heads that are diagonal and off-diagonal have very high frequency usage. The qualitative difference can be contrasted with Figure 17 and Figure 18 in which this high frequency usage is very clearly not as present. In Figure 4 this perhaps is even more clear when looking at a single layer: the two heads that mostly use the highest frequencies are heads 5 and 8 which correspond to a diagonal and off-diagonal head respectively. *We also validated this behaviour in Llama in our new revision in Figure 11*. We hope that this new figure in particular can help to convince you that this is indeed a wide-spread mechanism. We are happy to provide more evidence if you believe more is necessary, but we believe there is already a sufficient amount of evidence in the work (both theoretical and empirical).
>
> **Some prior works provide evidence that NoPE can still encode positional information [1][2]. So the fact that NoPE cannot learn a diagonal or off-diagonal pattern may not imply it cannot encode positions.**
>
> We agree with this point and in fact we do not make claims that NoPE is unable to encode positional information – we have clarified this in the paper. Our claim is instead that RoPE provides an *efficient* way to construct sharp diagonal or off-diagonal attention patterns (more generally positional attention patterns) through the use of the highest frequencies in particular. The difference between works such as [1, 2] is that we prove that specifically an attention head in isolation with NoPE is unable to learn these specific attention patterns, but we do not make any claims on what a deep model is able to learn. We however believe that studying what a single attention head can do is important because of an argument of efficiency: RoPE allows an attention head to implement something that would instead require more than one layer or component to achieve otherwise without it.
>
> **This claim comes from an analysis on a 2-dimensional case. However, it is possible that the 2D case is too restrictive and a higher dimensional case could have a different result.**
>
> We believe this 2-dimensional case to be quite applicable because as we show in our experiments, we find that the large norms often focus on distinct 2-dimensional bands. As such, we have indicative evidence that our theorem applies in the practical situations we study over several potent pre-trained LLMs. Of course, proving the more general case would be very interesting, but we believe this to be mathematically quite challenging – although we do not have reason to believe this to be untrue in the general case.

---

> > ### Author Response · Authors · 2024-11-18
> >
> > **Is your p-RoPE having a similar flavor as the partial rotary?**
> >
> > We thank you for pointing out this github issue, we have added this to our work to show that there have been similar ideas. We were unaware of this at the time of writing the paper. p-RoPE is indeed similar in nature to applying this to a part of the embedding, but of course is still different as the partial rotary encoding will still keep the lowest frequencies. We are very happy however to see that this has been tried already as this validates our findings.
> >
> > We believe that our work however importantly explains why p-RoPE or partial rotary encodings can work, allowing for the creation of robust semantic channels. We finally note that our proposal of p-RoPE is a small part of our overall work and in our opinion just a clear consequence of our findings.
> >
> > We thank you very much for your interesting points. We hope that you appreciate the additional ablations and results in the general comment and that our replies have helped to clarify your doubts. As you are the reviewer with the lowest score for our work, we would truly appreciate it if you could reconsider your opinion of our work given our comments and additional experiments and discussions. We hope that you agree with us that this work provides interesting and novel insights of RoPE, which would be very valuable for the community as a whole. We are of course very happy to answer any further questions.

---

> > > ### Author Response · Authors · 2024-11-24
> > > **Additional ablation**
> > >
> > > We wanted to also kindly add that due to our discussion with Reviewer T6HH, we have now added an experimental comparison to partial RoPE in Table 2 and expanded our discussion in the Appendix (Section D) on the similarities and differences. We also for convenience copy here the relevant discussion. We are happy to note that the reviewer has now decided to raise the score as well in support of the paper. We hope that you may find this relevant to your initial comments.
> > >
> > > **In response to Reviewer T6HH to a request of comparison with partial RoPE:**
> > >
> > > "We have now expanded our section on partial RoPE with a discussion on the differences! We expect partial RoPE and p-RoPE to achieve in different ways the same end goal. We believe our work provides a much more solid understanding on why techniques such as partial RoPE are helpful. We are happy to see techniques similar to p-RoPE being used already!
> > >
> > > For completeness, we have now also added an ablation in Table 2 with partial-RoPE showing that p-RoPE seems to show stronger performance. We hope that you can appreciate this ablation."

---

> ### Comment · Reviewer_DtVE · 2024-11-26
> **Post-rebuttal update**
>
> I appreciate the authors' thoughtful responses and the additional efforts put into conducting more ablation studies. After reviewing them, I believe these enhance the quality of the paper. Therefore, I have raised my score.

---

> > ### Author Response · Authors · 2024-11-26
> > **Thank you!**
> >
> > Dear Reviewer DtVE,
> >
> > Thank you for acknowledging our efforts! We are very happy that you found our responses useful and have upgraded your score.
> >
> > Could you please let us know what would be required, in your opinion, to bring the paper over the bar of acceptance?
> > We still have ~1.5 days to make concrete revisions to the paper, and we'd really like to try to make it happen!
> >
> > Best,
> > Authors

---

> > > ### Author Response · Authors · 2024-11-30
> > >
> > > Dear Reviewer DtVE,
> > >
> > > We thank you once again for your engagement and for already increasing your score.
> > >
> > > As the rebuttal period is now coming near an end, we would be really interested in hearing if you believe there are outstanding unresolved points in our work. We would really love to have a chance to address them!
> > >
> > > Best,
> > > Authors

---

### Official Review · Reviewer_7eRV · 2024-10-31

**Soundness:** 2
**Presentation:** 3
**Contribution:** 1
**Rating:** 5
**Confidence:** 4

**Summary:**

The paper provides practical insights into positional encoding for decoder-only models. However, further investigation is needed to establish the effectiveness and reliability of the theories presented.

**Strengths:**

Pros:

1. The paper offers a fresh discovery of Rotary Positional Encoding (RoPE), challenging the belief that it primarily helps decay attention coefficients with distance.

2. The paper gives an analysis of high and low RoPE frequencies, their roles in positional and semantic attention, and an innovative RoPE modification (p-RoPE) that demonstrates improvements in some cases.

3. Every section has a summary part which makes the whole paper clear and ready to read.

**Weaknesses:**

Cons:

1. The figures in the paper are somewhat confusing. For instance, while the paper emphasizes frequency aspects, Figure 1 illustrates vectors with identical frequency differences, which does not fully align with the paper’s focus. Additionally, the results are primarily presented using heat maps, which may appear monotonous and lack of expressiveness. Given that this is a language modeling task, presenting some results in natural language format would enhance readability and interpretability.

2. (MAIN LIMITATION) The experiments are limited to decoder-only models, making it unclear whether the findings can generalize to other transformer architectures. Whether the observed improvements stem from specific model structures (e.g., encoder-only or encoder-decoder models) rather than from a comprehensive enhancement applicable to diverse positional encodings of transformer. Additionally, the authors did not consider the impact of long-range dependencies or how the choice of positional encoding influences masked attention performance of decoder-only model. It would be valuable if the authors could report comparative results using unmasked attention (e.g., with encoder-only transformers) to examine these effects.

3. (MAIN LIMITATION) The proposed modification p-RoPE appears to primarily integrate elements of RoPE and NoPE, which may lack novelty. While the study advances understanding of RoPE, its applicability to other positional encoding methods, such as Alibi, remains limited, potentially constraining its relevance for models with alternative encoding schemes. The authors should consider incorporating other positional encoding methods and propose more innovative improvements to enhance the usage of lower and higher frequencies.

4. The paper states that higher frequencies correspond to positional attention, while lower frequencies correspond to semantic attention. However, the authors provide an ablation study only for the lower frequencies, without addressing both low and high frequencies. This aspect is insufficiently explained in the ablation experiments.

**Questions:**

Why did the authors provide an ablation study only for the lower frequencies rather than for both low and high frequencies?

---

> ### Author Response · Authors · 2024-11-18
>
> We thank you for your very interesting questions. We would like to address your points in full.
>
> **Figure 1 illustrates vectors with identical frequency differences, which does not fully align with the paper’s focus[...] heat maps, which may appear monotonous and lack of expressivenes**
>
> We believe that Figure 1 illustrates nicely one of the main mechanisms which we present in Proposition 3.1 and later use in Theorem 5.3. The figure depicts a single frequency as we believe that visualising multiple frequencies might be too cluttered and detract from the main point of the construction. To address this point we have amended the caption to clarify that the figure is focusing on a single frequency of RoPE as we agree with you that this might otherwise not be clear.
>
> Regarding the heat maps, we believe that they offer a very clear way to visualise our arguments. We also would like to point out that these types of heatmaps are very common in these types of visualisations [1, 2]. As we are mostly interested in visualising activations when we *take a mean* over different sequences, there is often no clear 1:1 correspondence to natural language. We would be happy to accommodate your request and modify our visualisations if you could point out which specific figure(s) you are referring to and how you would improve their message.
>
> **The experiments are limited to decoder-only models...**
>
> In this work we focus on Large Language Models such as Gemma and Llama that generate text in an auto-regressive manner. These language models do not use encoder-Transformers. As such, it is out of scope of this work to study encoder Transformers. We are further unaware of open-sourced and pre-trained LLMs that use an encoder architecture and use RoPE. We hope that given our comments convince you that this is not in fact a limitation of our work. Finally, it is also quite common for works to focus solely on studying decoder Transformers e.g. [3, 4]. We have made this more clear in the text.
>
> **...While the study advances understanding of RoPE, its applicability to other positional encoding methods, such as Alibi, remains limited, potentially constraining its relevance for models with alternative encoding schemes...**
>
> We are happy to read that you believe our study advances the understanding of RoPE. This is in fact the main goal of our work. As far as we are aware RoPE is by far the most popular positional encoding used in the training of autoregressive LLMs today – used in Gemma, Llama, and many others. For this reason we choose to focus only on understanding RoPE in this paper. Alibi is definitely a very interesting technique, but as far as we are aware much less used in frontier LLMs and mathematically very different. For these reasons, Alibi falls outside the scope of this work. We do however comment in the Appendix (Section B.2) on how we believe our work applies to different positional encodings.
>
> **The paper states that higher frequencies correspond to positional attention, while lower frequencies correspond to semantic attention. However, the authors provide an ablation study only for the lower frequencies, without addressing both low and high frequencies. This aspect is insufficiently explained in the ablation experiments.**
>
> We thank you for the interesting suggestion. We have added this as an ablation in Table 2 and are happy to report that removing the lowest frequencies indeed seems to outperform by a significant margin the removal of the highest frequencies. We hope that you find this result interesting.
>
> We sincerely thank you for your review and valuable comments. We hope that in light of our revisions (see general comment) and our response and additional ablations that you agree that our work has now improved. We would be grateful if you could consider upgrading your score under this new light. We are more than happy to answer any further questions during the rebuttal period.
>
> [1] Randomized Positional Encodings Boost Length Generalization of Transformers. ACL 2023.
>
> [2] Penzai + Treescope: A Toolkit for Interpreting, Visualizing, and Editing Models As Data. Johnson. Arxiv 2024
>
> [3] Transformers need glasses! Information over-squashing in language tasks. NeurIPS 2024. Barbero et al.
>
> [4] The expressive power of Transformers with Chain of Thought. ICLR 2024. Merrill et al.

---

> > ### Author Response · Authors · 2024-11-30
> >
> > Dear Reviewer 7eRV,
> >
> > We are very grateful for you review and insights on our paper. As the rebuttal period is coming to an end, we would really appreciate if you could let us know if you our responses and additional ablations have helped to improve your opinion on our work?
> >
> > We are available for further discussion at any point.
> >
> > Best,
> > Authors

---

### Official Review · Reviewer_rv9e · 2024-11-01

**Soundness:** 3
**Presentation:** 3
**Contribution:** 4
**Rating:** 8
**Confidence:** 4

**Summary:**

Authors explore how transformer models use RoPE frequencies to learn semantic and positional information. Authors provide a theoretical proof that RoPE does not force the decay in attention coefficient with distance, but instead can create specific patterns. In their experimental study of the Gemma 7B model, authors show that transformers mostly rely on low frequencies, while some heads display high frequency bands, mostly in 1st and last layers of the model. Authors further show that high-frequencies in RoPE provide a mechanism to encode positional information. Low frequencies are used as information channels that are not robust over long context. Finally, authors propose p-RoPE encoding that cuts low frequencies and can help to improve model's performance.

**Strengths:**

- The authors conduct a novel theoretical and empirical study of RoPE encodings in transformer models.
- They provide detailed proofs of their main claims.
- The paper is clear and well-written.
- This study can help researchers better understand the underlying mechanisms of popular transformer architectures and encourage research into alternative improved solutions.

**Weaknesses:**

- The empirical study is limited to a single Gemma architecture. While this is unlikely, some results may be artifacts of the specific model selected.
- In Section 3 authors train 2B model and show improvements on validation perplexity. While these results are positive, perplexity improvements do not always results in overall improvements in model's abilities. Authors could provide evaluation results on popular benchmarks* to build more convincing picture.


* see, for ex, Section 2.3 in https://arxiv.org/pdf/2307.09288

**Questions:**

1. Pre-training model from scratch is expensive and not always feasible. I was wondering if you can instead continuously train other (Gemma-7B, Llama-3 1/3/8B, etc) models for fewer steps but with p-RoPE approach? Do you think it would work or no and why?

2. Please, correct me if I'm wrong, but in the proof of Proposition 3.1:
for k>1, g_k=theta^{-2(k-1)/d} is not necessarily rational, but algebraic. Therefore, Lemma A.1 should be stated not for rational g, but for algebraic g, and it should use the fact that pi is transcendental.
line 680, cos(j-i-r) -> cos(j-i+r)
line 680, the comma in the displayed equation should be a period.
line 684, cos(j-i-r) -> cos(j-i+r)

---

> ### Author Response · Authors · 2024-11-18
>
> We are very happy to see you have enjoyed our work. We are particularly happy to read that you believe “this study can help researchers better understand the mechanism of popular transformer architectures” – which is exactly the point of this paper.
>
> **The empirical study is limited to a single Gemma architecture. While this is unlikely, some results may be artifacts of the specific model selected.**
>
> We thank you for this comment. We are excited to have now added results with Llama3.1 8B which show similar patterns. Interestingly, Llama3.1 8B has a 500k wavelength parameter and Grouped Query Attention. We find that the findings are very similar to Gemma, helping to support the generality of our work. The results can be found in the Appendix (Section C).
>
> **In Section 3 authors train 2B model and show improvements on validation perplexity. While these results are positive, perplexity improvements do not always results in overall improvements in model's abilities. Authors could provide evaluation results on popular benchmarks to build more convincing picture.**
>
> We have spent some time to add additional ablations to answer other reviewers which we hope you will find valuable. We agree that our evaluation of p-RoPE could be more extensive. The focus of the paper was not that of proposing a new type of positional encoding, but rather that of understanding RoPE. We found that our analysis immediately translated to improvements such as p-RoPE, but we saw p-RoPE more as an interesting ablation to verify our intuition. We are of course happy to see that you have already pointed out that our work helps to better understand RoPE.
>
> **Pre-training model from scratch is expensive and not always feasible. I was wondering if you can instead continuously train other (Gemma-7B, Llama-3 1/3/8B, etc) models for fewer steps but with p-RoPE approach? Do you think it would work or no and why?**
>
> This is a very interesting point and something which we found to be possible – although we found that performance was better when training from scratch. We believe it is possible as cutting off the lowest frequency rotations provides the “least amount of change” when re-computing the activations.
>
> **Please, correct me if I'm wrong, but in the proof of Proposition 3.1: for k>1, g_k=theta^{-2(k-1)/d} is not necessarily rational, but algebraic. Therefore, Lemma A.1 should be stated not for rational g, but for algebraic g, and it should use the fact that pi is transcendental. line 680, cos(j-i-r) -> cos(j-i+r) line 680, the comma in the displayed equation should be a period. line 684, cos(j-i-r) -> cos(j-i+r)**
>
> Yes you are exactly right. Thanks for catching this and for carefully checking our proofs! We have corrected this.
>
> We thank you again for endorsing our work and we hope that our additional results on Llama help to increase your confidence in the generality of our results. We are of course more than happy to answer any further questions.

---

> > ### Comment · Reviewer_rv9e · 2024-11-19
> > **Thank you for clarifications**
> >
> > Thank you for clarifications. >> "something which we found to be possible – although we found that performance was better when training from scratch" - was it included in the paper? If the evaluation results were not so good, it is still an interesting point for practical usage.
> >
> > I went over other reviews, authors' answers, and changes made in the paper. I think paper makes a solid contribution and has sufficient evidence to support main claims. I feel comfortable keeping my score at Accept.

---

> > > ### Author Response · Authors · 2024-11-23
> > > **Thank you for supporting our work!**
> > >
> > > We are happy to hear that you wish to maintain your score and also would like to thank you for your fast reply!
> > >
> > > **was it included in the paper? If the evaluation results were not so good, it is still an interesting point for practical usage.**
> > >
> > > We have not included it as it was preliminary experimentation, however we will include it in an eventual camera ready in the Appendix, as we believe we lack sufficient time in this rebuttal period to experiment with this appropriately.
> > >
> > > We once again thank you for your efforts in reviewing our work and for your positive score.

---

### Official Review · Reviewer_eLgp · 2024-11-05

**Soundness:** 2
**Presentation:** 3
**Contribution:** 3
**Rating:** 5
**Confidence:** 4

**Summary:**

This paper investigates the role of Rotary Positional Encodings (RoPE) in Transformer-based Large Language Models (LLMs). The authors challenge the common belief that RoPE's usefulness comes from its ability to decay token dependency with increasing relative distance. Instead, they explore how different frequencies in RoPE are utilized, particularly within the Gemma 7B model. The paper provides both theoretical and empirical analyses, proposes a modified RoPE, and highlights the importance of understanding positional encodings for scaling LLMs.

**Strengths:**

1. The paper provides a fresh perspective on RoPE, questioning existing assumptions and offering new explanations for its effectiveness.
2. The authors present mathematical proofs to support their claims, enhancing the credibility of their findings.
3. The use of the Gemma 7B model for empirical analysis adds practical relevance to the theoretical insights.

**Weaknesses:**

1. Although the observed phenomena and mathematical proofs can support the paper's point of view, the experimental performance does not seem good enough. The paper hopes to adapt to any context length, but the actual experimental results only have one result on 8K. And the evaluation of PPL is not comprehensive enough.
2. At the semantic level, the results of the models in Table 2 should be compared on the general benchmark or other tasks that are more representative of semantics, which will be more convincing.
3. At the positional level, it should be compared with similar experiments such as needle in a haystack or Ruler on long contexts to prove its long context expansion ability.

**Questions:**

1. I have some doubts about the display of Figure 2. Normally, for the same qk, different relative distances should have a gradually decreasing effect. But does different qk introduce different variables, making the comparison unfair?
2. Is there any empirical experiment on how many frequencies are cut off for the best effect?
3. Will removing low-frequency RoPE make the effect on short text worse?

---

> ### Author Response · Authors · 2024-11-18
>
> Thanks for your excellent points! We would like to address your comments below.
>
> **The paper hopes to adapt to any context length, but the actual experimental results only have one result on 8K. And the evaluation of PPL is not comprehensive enough. ...**
>
> Thanks so much for the great points regarding how to improve our experimental evaluation. We invested time in improving our evaluation of p-RoPE and have added 2 new ablations, which have also been requested by other reviewers.
>
> We completely agree with your point that the experimental evaluation could be more extensive, but we would like to stress that the point of this work is not that of proposing a new positional encoding, but rather to better understand RoPE. As such we saw the proposal of p-RoPE as a way great way to validate the intuition we develop in our paper. We believe that the 2 new ablations we provide help to support such a claim.
>
> In other words, while we offer p-RoPE as a practical solution, this is not the main focus of the paper and is only there to validate experimentally our reasoning in Section 6 that the low frequency channels could be removed as they are not robust. We however also believe that our improved experimental section is still rather interesting as we train 2B parameter Gemma models, showing improvements in perplexity. Of course we agree with you that perplexity has its limitations as well and we have covered this in the Appendix (Section B.5).
>
> We are happy to overall see that you seem to appreciate our contributions towards the understanding of RoPE and kindly request you to evaluate our work on whether it provides a better understanding and novel insights on why RoPE is useful. We believe that addressing common misconceptions such as the claim that RoPE helps to decay activations with distance to be important – as pointed out also for example by Reviewer rv9e in their strengths. Many works tend to in fact propose new positional encodings, but very few if any attempt to truly understand why the existing widespread positional encodings we have to date work so well.
>
> **I have some doubts about the display of Figure 2. Normally, for the same qk, different relative distances should have a gradually decreasing effect. But does different qk introduce different variables, making the comparison unfair?**
>
> We thank you for the great point. We agree with your perspective and we have added in the Appendix (Section B.4 – see Figure 9) another synthetic experiment where the queries and keys are “constant” from a Gaussian – i.e. we sample a *single* query and a *single* key from a Gaussian and then repeat them up to the sequence length. It is clear also in this case that there is no clear decay. We hope that this addresses your point.
>
> **Is there any empirical experiment on how many frequencies are cut off for the best effect?**
>
> In our experiments, we found a value of 25% cutoff (0.75-RoPE) to be the best performing. We report in Table 2 results for a cutoff of 0.25 and 0.75, of course we believe that a finer grid search is likely to yield even better results. Please also compare our ablation with 0.75-RoPE_{reversed} in Table 2, where we cut-off the highest frequencies. We show that this does not work as well, aligning with the intuition derived in our work.
>
> **Will removing low-frequency RoPE make the effect on short text worse?**
>
> Thanks for the interesting question. We have no reason to expect shorter texts to fare worse given our experiments. The datasets we use include a number of short documents with less than $1,000$ tokens.
>
> We would like to thank you for your excellent comments. As most focused on our p-RoPE experiments, we hope that you could appreciate that we believe most of the contributions are actually towards the understanding of RoPE, rather than the proposal of a new type of PE. In this light, we hope that, with our new ablations and the additions we mention in our global comment, that you kindly consider upgrading your score. We are more than happy to keep engaging if you have further questions.

---

> > ### Comment · Reviewer_eLgp · 2024-11-25
> >
> > I appreciate the authors' response. While the rebuttal addressed some questions, I still have concerns about the experimental settings and results.
> >
> > 1. The effect of Figure 9 is indeed shocking. But what does the phrase "perfectly aligned" mean? In other words, what is the difference between the settings of Figure 9 and Figure 2 (a)?
> > 2. The author's description of semantics and position in Table 1 is not experimentally verified.
> > 3. Even though the author says the focus is on understanding, I still agree with the theoretical proof in the paper, but only PPL seems less convincing in the experiment.

---

> > > ### Author Response · Authors · 2024-11-25
> > >
> > > We would like to thank you very much for your response and for your efforts as a reviewer. We are happy to have addressed some of your questions. We would like to address your remaining points.
> > >
> > > **The effect of Figure 9 is indeed shocking. But what does the phrase "perfectly aligned" mean? In other words, what is the difference between the settings of Figure 9 and Figure 2 (a)?**
> > >
> > > We thank you for the question. To clarify:
> > >
> > > *Figure 2 (a)*: the queries and keys are repeated vectors of all-ones. In other words, we take the first queries and keys q_1 and k_1 as all-ones vectors and repeat them n times.
> > >
> > > *Figure 9*:  We sample 1 query and 1 key from a Gaussian distribution, i.e. the queries and keys are different vectors now. We then repeat the query n times and the key n times. We then show the effect of RoPE on this repeated sequence.
> > >
> > > The difference is that in Figure 2 (a) they are always perfectly aligned as the base vectors are all vectors of 1s. Since they are the same vector, their initial angle between them is always 0 (what we mean by perfectly aligned). Instead, in Figure 9, while the queries and keys are repeated, they are *different* Gaussian random vectors, so they are not the same vector. We see that therefore the vectors being repeated is not sufficient, but they also have to be aligned (the same vector) for them to decay.
> > >
> > > Just for completeness, the difference in Figure 2 (b) is that here we sample Gaussian vectors for each query and key, so we sample n queries and n keys from a Gaussian, while in Figure 9, we sample only 1 query and 1 key and then repeat them n times.
> > >
> > > Please let us know if this is now hopefully more clear!
> > >
> > > **The author's description of semantics and position in Table 1 is not experimentally verified.**
> > >
> > > We agree with you and have clarified in the table that these are the discussed *theoretical* properties. Thanks for pointing this out!
> > >
> > > **Even though the author says the focus is on understanding, I still agree with the theoretical proof in the paper, but only PPL seems less convincing in the experiment.**
> > >
> > > We are happy that you agree with the proofs in our paper! There are many works that propose a new type of positional encoding, but we believe that it is also very valuable to improve our understanding of the positional encodings being used today. We hope that you agree that our paper indeed does contribute to a better understanding of RoPE. In fact, we are happy that in your review you mention that we “question existing assumptions about RoPE”  and that our analysis has “practical relevance” which we believe is really the main point of this paper! We hope that this work can in fact provide the community with a more solid understanding of why RoPE is useful.
> > >
> > > The experimental evaluation of our p-RoPE method is mostly to validate our intuitions derived in the sections of this paper. We hope that the 3 added experimental ablations help to solidify to you that p-RoPE is an interesting and valid approach. We completely agree that we could experiment on more tasks, but we believe this to be somewhat outside the main scope of this work which is that of understanding RoPE and not the proposal of a new PE.
> > >
> > > We once again thank you for your review and are happy to answer any further questions.

---

> > > > ### Author Response · Authors · 2024-11-30
> > > >
> > > > Dear Reviewer eLgp,
> > > >
> > > > We thank you once again for your engagement with us and your further questions!
> > > >
> > > > We are wondering if our response has addressed your remaining concerns? As we the rebuttal period is almost over, we would really be happy to have the chance to address any final questions before we cannot anymore.
> > > >
> > > > Best,
> > > > Authors

---

### Official Review · Reviewer_T6HH · 2024-11-05

**Soundness:** 3
**Presentation:** 3
**Contribution:** 3
**Rating:** 8
**Confidence:** 4

**Summary:**

This paper delves into the role of Position Embedding (PE) in LLMs, challenging the traditional view that RoPE primarily attenuates attention weights as the relative distance between words increases. It proposes a new hypothesis that RoPE constructs position-attention patterns (e.g., diagonal or previous-token focus) using high-frequency components while leveraging low frequencies to convey semantic information. Several case studies and theoretical analyses support this hypothesis.

Overall, the paper provides insightful findings and hypotheses on a key component of LLMs: position encoding. However, the arguments mainly rely on case studies with gemma-7B, and the experiments lack diversity in the foundation models (e.g., missing LLaMA series) and tasks (e.g., language model vs. QA vs. code...). Additionally, some parts of the proofs contain errors.

**Strengths:**

1. This paper successfully challenges, both theoretically and empirically, the traditional view that RoPE attenuates attention weights as relative distance between tokens increases.
2. The authors' hypothesis about the roles of the high-frequency and low-frequency components of RoPE is novel and insightful.

**Weaknesses:**

1. The experimental validation lacks diversity in both foundation models and datasets.
2. Some perspectives and proofs regarding NoPE contain errors, while they don't affect the main conclusion, they may mislead readers.
3. The discussion of related work is insufficiently thorough, and few papers are cited (only about one page).
4. Throughout Section 4, the authors conceal a core assumption: that attention scores are interpretable or meaningful. Higher attention scores for certain tokens imply a meaningful preference in the model, giving special significance to the larger norms in Equation at Line 257. I want to point out that this assumption is still being debated, and I suggest that the authors make it explicit.
> Is Attention Explanation? An Introduction to the Debate (Bibal et al., ACL 2022)

**Questions:**

1. In practice, do the phenomena observed on gemma-7b apply to other LLM backbones, such as llama-2 or llama-3?
2. The phenomena observed in this paper are based on what scale and type of data? Could the findings be validated across various tasks, such as language modeling, code, or QA?
3. In Line 190, the authors claim that NoPE has strong OOD capabilities, but Kazemnejad et al., 2024 only validated this for sequences of length up to 50, where NoPE performed slightly better than other PEs, without exhibiting exceptionally strong OOD performance (refer to their Figure 3). The following works explored NoPE's extrapolation, and it can be seen that the generalization ability of the NoPE baseline is limited:
> Length Generalization of Causal Transformers without Position Encoding (Wang et al., Findings 2024)
>
> [Neurips24 spotlight] Exploring Context Window of Large Language Models via Decomposed Positional Vectors
4. In Line 497, the authors state, "p-RoPE is in spirit similar to the idea of increasing the wavelength of RoPE from 10,000 to 500,000," so I suggest that Table 2 should include RoPE with a base of 500,000.
5. How do the observations in Section 4 change across different RoPE variations, such as 0.25-RoPE, 0.75-RoPE, RoPE_10000, and RoPE_500000? For example, in Figure 3, how does the norm distribution of low frequencies vary across these models?
6. In Line 726, this proof only applies to the first layer of NoPE’s attention heads. Starting from the second layer, the assumption
$a_{3,3}=<q_3, k_3>=<q_3, k_2>=a_{3,2}$
no longer holds.
7. Attention heads with specific patterns have been widely studied. Besides the patterns discussed in this paper (e.g., diagonal or previous-token focus), could the authors observe and analyze other representative attention patterns, such as special token focus, punctuation focus, and locality focus? Please refer to typical patterns in the following paper:
> [ICLR24] Model Tells You What to Discard: Adaptive KV Cache Compression for LLMs

---

> ### Author Response · Authors · 2024-11-18
>
> Thank you so much for your thorough review and the great suggestions!
>
> We address W3 and W4 first as W1 and W2 are also repeated in the questions.
>
> **W3 The discussion of related work is insufficiently thorough**
>
> We agree with your point that our related work discussion was a bit limited. We have added a new section in the Appendix (Section B.1) with a pointer in the main text covering a large number of works. We are of course happy to include works you believe we may have missed and hope that our additional literature review addresses your concern.
>
> **W4 ... the authors conceal a core assumption: that attention scores are interpretable or meaningful.  I want to point out that this assumption is still being debated, and I suggest that the authors make it explicit ...**
>
> We thank you for the additional reference, we have added this as a note in the main text.
>
> **Q1 In practice, do the phenomena observed on gemma-7b apply to other LLM backbones, such as llama-2 or llama-3?**
>
> Thanks for the great comment, we agree that this was missing in the original version of the manuscript. We are excited to have added new results on Llama3.1 8B, showing similar results to Gemma. This new section can be found in the Appendix (Section C). Importantly the Llama model uses a different wavelength of 500k and grouped query attention – leading to very interesting insights which we discuss in the appendix and allowing us to further support our claims. In particular, we find that Llama still prefers the lower frequencies, but now leverages a greater spread of them due to the increased max wavelength. We also find diagonal attention patterns being constructed through the highest frequencies. We hope that you find these new additions valuable.
>
> **Q2 The phenomena observed in this paper are based on what scale and type of data? Could the findings be validated across various tasks, such as language modeling, code, or QA?**
>
> Our investigations rely on the publicly available pretrained Gemma models (and now also Llama) that are trained on a very large corpus. We believe the details of the training data are not public but for instance in Gemma they report “data from web documents, mathematics, and code” [1]. Llama is also trained on “5% multilingual data” [2]. As such we believe our findings to be rather broad due to the scale, breadth, and amount of training present in these models.
>
> **Q3 In Line 190, the authors claim that NoPE has strong OOD capabilities, but Kazemnejad et al., 2024 only validated this for sequences of length up to 50...**
>
> We thank you for the additional references, we have now included these as a disclaimer in the main text. We would like to clarify that we do not make claims about NoPE being the best method to generalise to OOD, but simply that NoPE provides a mechanism to construct certain attention patterns in a way that is perfectly robust to relative distance – as by construction NoPE is invariant to relative distance. An example of this is attending to the BOS token robustly, which we have now added a more detailed discussion in the Appendix (Section E.1).
>
> **Q4 In Line 497, the authors state, "p-RoPE is in spirit similar to the idea of increasing the wavelength of RoPE from 10,000 to 500,000," so I suggest that Table 2 should include RoPE with a base of 500,000.**
>
> We agree that this ablation would be valuable in our work. We have now added the results with a base of 500k in Table 2 and are happy to report that p-RoPE outperforms this baseline as well.
>
> **Q5 How do the observations in Section 4 change across different RoPE variations, such as 0.25-RoPE, 0.75-RoPE, RoPE_10000, and RoPE_500000? For example, in Figure 3, how does the norm distribution of low frequencies vary across these models?**
>
> This is a very interesting question. With the new Llama model, we now are able to see the effect of increasing the wavelength to 500k. In particular, it is precisely what we predicted given our investigation. When comparing Figure 10 (Llama 500k wavelength) and Figure 13 (Gemma 10k wavelength), we see that Gemma is much more limited to the very lowest frequencies, while the increased wavelength allows the model to use many more frequencies in Llama. We have provided a much more detailed discussion in the Appendix (Section C) on why we believe this is the case. Overall, we believe these new results to heavily back up our findings and thank you for the suggestion on this comparison.
>
> Due to logistical reasons, it is hard for us to perform the same analysis on the models we trained ourselves, but we hope that you can appreciate the comparison we have now added between Gemma at 10k and Llama at 500k. We also believe that these results are likely to be more interesting as these models have of course been trained for much longer.

---

> > ### Author Response · Authors · 2024-11-18
> >
> > **Q6 In Line 726, this proof only applies to the first layer of NoPE’s attention heads. Starting from the second layer, the assumption  a3,3=<q3,k3>=<q3,k2>=a3,2 no longer holds.**
> >
> > We thank you for checking our proof. We would like to clarify that this is not an error in our proof. We specifically study what can be implemented in a *single* attention head in isolation and thus in a single layer. The motivation for this is that we care about studying the “efficiency” of a single attention head – i.e. what can a single head by itself implement. For instance, in Gemma 7B one can find previous-token or diagonal heads immediately in the first layer, and our proof shows that these heads would be impossible to implement if one simply used NoPE. This means that RoPE provides additional mechanisms over NoPE to construct heads that the Transformer is finding useful during the learning process. We have made this more clear in the paper as we appreciate that this detail might be missed.
> >
> > **Q7 Attention heads with specific patterns have been widely studied. Besides the patterns discussed in this paper (e.g., diagonal or previous-token focus), could the authors observe and analyze other representative attention patterns...**
> >
> > We thank you for the additional reference, which we now cite. We have added a more detailed analysis (Appendix, Section E.1) of the Apostrophe head which is both a “punctuation head” and a “special token head” as it either attends to the BOS “special” token or a punctuation token. We believe this analysis to be rather interesting and likely the first of its kind: reverse engineering the mechanism through which RoPE allows the head to be constructed.
> >
> > Further, we would like to point to Figure 18 for an example of a different type of head not added in the main text. We found this to be rather representative of a number of heads, with the high frequencies relatively inactive and “high norm” low frequency bands. We are happy to include more examples if you believe this would be useful to the work, but we believe to already have a large breadth of examples and would prefer to avoid adding too many, in order to preserve clarity and clarity.
> >
> > We would like to sincerely thank you for your very thorough review and the excellent suggestions, which we believe have helped to strengthen our work. We would be grateful if you could consider upgrading your score to our work if you are satisfied with our answers. Of course we are very happy to answer any further questions.
> >
> > [1] Gemma: Open Models Based on Gemini Research and Technology. Gemma Team, 2024.
> >
> > [2] The Llama 3 Herd of Models. Meta, 2024.

---

> > > ### Comment · Reviewer_T6HH · 2024-11-21
> > > **Thanks for the rebuttal and the detailed responses.**
> > >
> > > I appreciate the authors' effort during the rebuttal phase in conducting additional experiments and making modifications to the descriptions. Most of my concerns have been resolved. Considering the resolution of key issues (e.g., the use of the LLaMA series as the base, additional baselines, and more analyses of attention patterns), I will raise my score.
> > >
> > > However, there are still three remaining concerns that I would like to address:
> > >
> > > 1. The authors seem to have misunderstood my Question 2, which is about the dataset on which these analyses or statistical experiments (e.g., 2-norm) are conducted, not about the pretraining dataset of Gemma.
> > >
> > > 2. In response to Question 3, the authors claimed, *"as by construction NoPE is invariant to relative distance."* I would like to point out that this understanding of NoPE is incomplete. NoPE simply does not provide explicit PE information to the Transformer, but it can implicitly learn both absolute and relative positional information through the causal mask. For example, [1] has demonstrated that a single-layer NoPE can learn absolute positional information, and a two-layer NoPE can learn relative positional information. Furthermore, Kazemnejad et al., 2024, showed through similarity analysis of hidden representations that NoPE's representations are highly similar to those of T5. Likewise, the Proposition 5.3 in this paper merely highlights the problem with a single-layer NoPE. However, in practice, single-layer Transformer-NoPE is almost never used, so this conclusion is not particularly exciting.
> > > [1] *Latent Positional Information is in the Self-Attention Variance of Transformer Language Models Without Positional Embeddings*, ACL2023 Honorable Mentions.
> > >
> > > 3. Finally, I would like to discuss p-RoPE and partial-RoPE (as suggested by reviewer DtVE). Let’s assume \(d_{head} = 128\):
> > >    - The original RoPE uses `inv_freq = 1.0 / (base ** (torch.arange(0, 128, 2) / 128))`, with the lowest frequency being $1.0 / \text{base}^{126/128}$.
> > >    - 0.5-RoPE uses `inv_freq = 1.0 / (base ** (torch.arange(0, 64, 2) / 128))`, with the lowest frequency being $1.0 / \text{base}^{62/128}$.
> > >    - Partial-RoPE (50% part) uses `inv_freq = 1.0 / (base ** (torch.arange(0, 64, 2) / 64))`, with the lowest frequency being $1.0 / \text{base}^{62/64}$.
> > >
> > >    Clearly, 0.5-RoPE removes the low-frequency components of positional encoding more significantly, but a comparison between p-RoPE and partial-RoPE remains valuable (if more time and resources are available). The low-frequency components in the original RoPE are effective at learning semantics because these dimensions are less sensitive to positional changes, which is beneficial for semantic learning. In partial-RoPE, while the low-frequency information still exists, half of the dimensions are unaffected by PE. This implies that semantic learning does not necessarily need to rely on low-frequency dimensions. I hope the authors can include this experiment in the next version.
> > >
> > >    Additionally, regarding partial-RoPE, beyond its mention in the GitHub issue, it has also been used in GPT-NeoX and DeepSeek-V2 (which can serve as formal references).

---

> > > > ### Author Response · Authors · 2024-11-23
> > > > **Thank you for supporting our work and the great review!**
> > > >
> > > > Thanks for your reply! We are happy to hear that you intend to raise your score. We would like to also thank you for the follow up questions and for your engagement with our work.
> > > >
> > > > **The authors seem to have misunderstood my Question 2, which is about the dataset on which these analyses or statistical experiments (e.g., 2-norm) are conducted, not about the pretraining dataset of Gemma.**
> > > >
> > > > We apologise as we have indeed misunderstood your original question, thanks for clarifying. This is a very interesting point and we completely agree that this would be an interesting ablation to conduct. We have added in the Appendix a new figure with a number of different domains: Italian, Chinese, Code, and Arithmetic. The results can be found in the newly added Figure 19. The patterns we find for these types of prompts are very similar across the different domains.
> > > >
> > > > **In response to Question 3, the authors claimed, "as by construction NoPE is invariant to relative distance." I would like to point out that this understanding of NoPE is incomplete. NoPE simply does not provide explicit PE information to the Transformer, but it can implicitly learn both absolute and relative positional information through the causal mask […]**
> > > >
> > > > We thank you for the very interesting comment. We would like to start by agreeing with you that we do not believe that NoPE is unable to learn positional information and perhaps our statement on “NoPE being invariant to relative distance” was not very precise. With that, we meant that over very long context even the slowest frequencies in RoPE will eventually destroy information – but that instead with NoPE, this semantic destruction does not occur because by construction the individual channels *can be used* in a way that is invariant to relative distance. For example, in our explanation of the apostrophe head, we show that a particular channel is being used to detect the BOS token, but that this mechanism will eventually break due to the rotations if the context is long enough. Instead, if this was a non-rotating channel, this mechanism would be much more robust. You are right in pointing out that this does not have to be true if the previous layer learns some positional artefacts, but our claim is more on the possibility of this to occur with NoPE and on the impossibility with RoPE.
> > > >
> > > > You make a great point when you mention that works such as [1] and Kazemnejad et al point out that indeed with NoPE the Transformer can still learn a positional bias – given more than one layer. We still believe that our result is interesting because it provides understanding at a different level. While it is definitely interesting to study the expressive power of a sequence of layers, we believe there is still value in studying mechanisms that can occur in a specific attention head. Not only because we can show that certain mechanisms would not be able to occur with NoPE at the first layer, but also because many of these compositional arguments from a practical perspective will be less robust and less efficient. For instance, the proof of Kazemnejad et al relies on the universal approximation theorem to map 1/t to some desired decay function, which in practice of course points to potential issues with generalisation for instance when t is out of distribution. We have also added a comment on this in the Appendix (in a paragraph at the end of Section A.2).
> > > >
> > > > We hope that this clarifies our position on this and are of course happy to keep discussing this with you.
> > > >
> > > > **Finally, I would like to discuss p-RoPE and partial-RoPE (as suggested by reviewer DtVE). […]**
> > > >
> > > > We thank you for the additional references – we have now expanded our section on partial RoPE with a discussion on the differences! We expect partial RoPE and p-RoPE to achieve in different ways the same end goal. We believe our work provides a much more solid understanding on why techniques such as partial RoPE are helpful. We are happy to see techniques similar to p-RoPE being used already!
> > > >
> > > > For completeness, we have now also added an ablation in Table 2 with partial-RoPE showing that p-RoPE seems to show stronger performance. We hope that you can appreciate this ablation.
> > > >
> > > > Overall, we sincerely thank you for the effort you have put in reviewing our manuscript and thank you again for mentioning that you wish to increase your score. We believe your comments have been really useful in improving our work and welcome any more you may have!

---

> ### Comment · Reviewer_T6HH · 2024-11-24
> **Thanks for the rebuttal.**
>
> Thank you for your detailed response. I have no further questions. I raised my score (from 5 to 8).

---

> > ### Author Response · Authors · 2024-11-24
> > **Thank you!**
> >
> > Thank you very much for your thorough review and your useful feedback. We found it very helpful. We are happy that you have decided to raise your score! Best, Authors.

---

### Author Response · Authors · 2024-11-18
**General Comment**

We would like to thank the reviewers for their efforts in assessing our work and the valuable feedback. We have accordingly made significant improvements following the suggestions and comments. For convenience, *we have marked in the revised PDF in blue* the added sections and the captions of newly added figures.

Below we summarise our changes:

- Llama results: Reviewers T6HH and rv9e requested inclusion of an additional model. We have **added results with Llama 3.1 8B** in the
Appendix (Section C) showcasing similar patterns to Gemma, which we believe greatly increases the value and generality of our claims.
- Added in Figure 19 an ablation of the usage of RoPE frequencies in different domains (Italian, Chinese, Code, and Arithmetic) to show that our results generalise over different types of domains.

- Added **additional experimental ablation** comparing p-RoPE to increasing the max wavelength parameter in Table 2 as requested by Reviewer T6HH. We are happy to report that p-RoPE outperforms this baseline.
- Added **additional experimental ablation** comparing p-RoPE to removing the *highest* frequency instead of the lowest ones as suggested in p-RoPE in Table 2, as requested by Reviewer 7eRV. We are happy to report that p-RoPE outperforms this baseline.
- Added **additional experimental ablation** comparing p-RoPE to partial-RoPE in Table 2, as requested by Reviewer 7eRV. We are happy to report that p-RoPE outperforms this baseline.

- Added a more **detailed analysis of the “apostrophe head”** in the Appendix (Section E.1)  – reverse engineering how RoPE is used to construct this particular head.
- **Additional synthetic result** showing that RoPE does not decay over a constant sequence of *repeated* queries and keys sampled *once* from a Gaussian as requested by Reviewer eLgp in Appendix B.4.
- **Improved related works**: Following the advice from Reviewer T6HH, we have added a more detailed discussion of related works in the Appendix (Section B).

We hope that our revisions have strengthened our contributions and would like to thank the reviewers for their valuable suggestions.  We look forward to productive rebuttal discussions.

---

### Public Comment · ~Mingyu_Xu1 · 2024-11-26
**An informal comment**

Hello, I'm glad to find this article on ICLR.

In line 423, "We therefore suspect that with larger and larger contexts, the $\theta$ base wavelength of RoPE will have to also be increased accordingly."

I think the following paper may get you,  "there is an absolute lower bound for the base value to obtain certain context length capability".[1]

Good luck!

[1]Base of RoPE Bounds Context Length https://openreview.net/forum?id=EiIelh2t7S

---

> ### Author Response · Authors · 2024-11-26
>
> Thanks for pointing this out, we have updated line 423 to reference this work.
>
> We are indeed happy to see evidence for our conjecture.
>
> Best,
> Authors

---

### Meta-Review · Area_Chair_RdPT · 2024-12-25

**Metareview:**

**Summary:** This paper provides a theoretical and empirical study of Rotary Positional Encodings (RoPE) in transformer-based LLMs. The authors challenge the conventional belief that RoPE facilitates token dependency decay with increasing distance. Instead, the work hypothesizes that RoPE's utility lies in its ability to construct robust positional attention patterns using high frequencies, while low frequencies encode semantic information. The paper validates these hypotheses through mathematical proofs and extensive experiments, including a novel modification of RoPE, named p-RoPE, which demonstrates performance improvements in specific settings. The study advances the community's understanding of positional encodings and proposes actionable insights for future model design.

**Decision:** The paper makes a significant theoretical and empirical contribution by deepening our understanding of RoPE and proposing meaningful modifications. Despite some concerns regarding the experimental breadth (e.g., limited evaluation on downstream tasks and reliance on perplexity as the primary metric), the added ablations and discussions during the rebuttal phase strengthened the case for the paper’s claims. Reviewer 7eRV raised concerns regarding novelty and limited to decoder-only models. However, based on the prevailing techniques in this field, I do not consider these to be weaknesses of this work.

Overall, the contributions of this paper outweigh its shortcomings, and I recommend its acceptance. I encourage the authors to incorporate the reviewers' feedback and the additional content provided in the rebuttal into the final version to further enhance the quality of the paper.

**Additional Comments On Reviewer Discussion:**

The discussion highlighted the paper's strong theoretical contributions and novel insights into RoPE mechanisms, which were well-supported by proofs and experiments. Reviewers appreciated the added ablations, such as analysis across different frequency ranges and results on Llama 3.1 8B, which demonstrated generality. However, some concerns about limited experimental diversity and reliance on perplexity as the primary metric for evaluation remained partially unresolved. Despite this, the reviewers largely agreed that the paper provides valuable understanding of positional encodings and is a significant contribution to the field.

---

### Decision · Program_Chairs · 2025-01-22

Accept (Poster)